# Platinum(IV) Prodrugs Incorporating an Indole-Based Derivative, 5-Benzyloxyindole-3-Acetic Acid in the Axial Position Exhibit Prominent Anticancer Activity

**DOI:** 10.3390/ijms25042181

**Published:** 2024-02-11

**Authors:** Angelico D. Aputen, Maria George Elias, Jayne Gilbert, Jennette A. Sakoff, Christopher P. Gordon, Kieran F. Scott, Janice R. Aldrich-Wright

**Affiliations:** 1School of Science, Western Sydney University, Sydney, NSW 2751, Australia; a.aputen@westernsydney.edu.au (A.D.A.); m.elias3@westernsydney.edu.au (M.G.E.); c.gordon@westernsydney.edu.au (C.P.G.); 2Ingham Institute, Sydney, NSW 2170, Australia; kieran.scott@westernsydney.edu.au; 3Calvary Mater Newcastle Hospital, Newcastle, NSW 2298, Australia; jayne.gilbert@newcastle.edu.au (J.G.); jennette.sakoff@newcastle.edu.au (J.A.S.); 4School of Medicine, Western Sydney University, Sydney, NSW 2751, Australia

**Keywords:** chemotherapy, platinum(II), cisplatin, 56ME*SS*, platinum(IV), cytotoxicity, ROS, mitochondria, HDAC

## Abstract

Kinetically inert platinum(IV) complexes are a chemical strategy to overcome the impediments of standard platinum(II) antineoplastic drugs like cisplatin, oxaliplatin and carboplatin. In this study, we reported the syntheses and structural characterisation of three platinum(IV) complexes that incorporate 5-benzyloxyindole-3-acetic acid, a bioactive ligand that integrates an indole pharmacophore. The purity and chemical structures of the resultant complexes, **P-5B3A**, **5-5B3A** and **56-5B3A** were confirmed via spectroscopic means. The complexes were evaluated for anticancer activity against multiple human cell lines. All complexes proved to be considerably more active than cisplatin, oxaliplatin and carboplatin in most cell lines tested. Remarkably, **56-5B3A** demonstrated the greatest anticancer activity, displaying GI_50_ values between 1.2 and 150 nM. Enhanced production of reactive oxygen species paired with the decline in mitochondrial activity as well as inhibition of histone deacetylase were also demonstrated by the complexes in HT29 colon cells.

## 1. Introduction

The classical cytotoxic platinum(II) drug, *cis*-diammineplatinum(II) dichloride (cisplatin) and its direct analogues, *trans*-*L*(1*R*,2*R*-diaminocyclohexane) oxalatoplatinum(II) (oxaliplatin) and *cis-*diammine(1,1-cyclobutanecarboxylato)platinum(II) (carboplatin) (Figure 1), are a class of square planar platinum(II) anticancer drugs, which have irrevocably evolved as working pillars of chemotherapy [1,2,3,4,5]. The potency of platinum(II) drugs is demonstrated through their ability to induce deoxyribonucleic acid (DNA) lesions by covalently binding to DNA [6]. Consequently, DNA replication and transcription are effectively ceased, which induces apoptosis [7]. It is also noteworthy that while DNA is the therapeutic target of these drugs, inevitably, they also exhibit non-specific binding with phospholipids, thiols, ribonucleic acid, and other biomolecules that contain nitrogen, sulfur, and oxygen atoms [8,9,10]. Unfortunately, this singularity reflects the non-selectivity of the drugs resulting in decreased therapeutic efficacy and increased incidence of pernicious side effects, which patients commonly experience when receiving platinum-based regimens [11,12,13,14,15]. Furthermore, drug resistance is another clinical limitation associated with the use of platinum(II) drugs, which often develops when treatment is prolonged [16,17]. Because of this, combinatorial treatment is employed, wherein therapeutic agents are administered together with platinum(II) drugs in an attempt to enhance treatment effectivity [18]. However, this type of regimen can complicate patient care and interfere with therapeutic outcomes since each co-administered agent has a unique pharmacokinetic index [19,20].

Cisplatin, oxaliplatin and carboplatin are effective anticancer agents. Regrettably, their disadvantageous traits proved to be burdensome to patients, and thus it is only ethical to develop safer chemotherapeutic alternatives that can improve cancer treatment experience. Many platinum-based complexes have been previously synthesised and investigated in the hope of finding another chemotherapeutic agent that is as potent as cisplatin, but without the toxic side effects, and has the potential to circumvent chemoresistance [21,22,23,24]. Some examples include zeniplatin [25], spiroplatin [26], enloplatin [27], aroplatin [28] and miboplatin [29] (Figure 2). These third-generation platinum(II) complexes were in clinical trials, but were eventually withdrawn either because of demonstrated severe toxicity or limited antitumour activity.

A collection of structurally related platinum(II) complexes have also been investigated and numerous studies have confirmed their impressive biological activity against multiple human cell lines [30,31,32,33,34,35]. The chemical structures of these complexes are represented as **[Pt^II^(H_L_)(A_L_)]^2+^**, where H_L_ is a bulky heterocyclic ligand (i.e., 1,10-phenanthroline (phen), 5-methyl-1,10-phenanthroline (5-Mephen) or 5,6-dimethyl-1,10-phenanthroline (5,6-Me_2_phen)) and A_L_ is a chiral ancillary ligand (i.e., 1*S*,2*S*-diaminocyclohexane (*SS*-DACH or DACH)). [Pt^II^(phen)(*SS*-DACH)]^2+^ (**PHEN*SS***), [Pt^II^(5-Mephen)(*SS*-DACH)]^2+^ (**5ME*SS***) and [Pt^II^(5,6-Me_2_phen)(*SS*-DACH)]^2+^ (**56ME*SS***) are three notable examples from this class (Figure 3). **56ME*SS*** has proven to be the most potent and has been reported to promote epigenomic changes by interfering with the cytoskeleton, cell-cycle proteins, mitochondria, and nuclear DNA [36,37]. Despite the remarkable in vitro activity of **56ME*SS***, preliminary in vivo studies indicated that the drug was ineffective in suppressing tumour on BD-IX rats with peritoneal carcinoma and caused nephrotoxicity [38]. Intriguingly, when **PHEN*SS*** was administered on mice with prostate tumour, no toxicity was detected and the rate of tumour regression was comparable to cisplatin [32]. To further elucidate this irregularity, the equivalent platinum(IV) derivatives of the complexes were synthesised and investigated, with preliminary evidence showing an enhancement in pharmacokinetic and pharmacological properties [39,40].

Researchers envisioned that designs based on kinetically inert octahedral platinum(IV) complexes would deliver safer, well-tolerated and effective anticancer treatment [41,42,43,44,45]. It is generally accepted that platinum(IV) complexes are the likely chemotherapeutic replacements for their equivalent platinum(II) parent complexes because of their kinetic inertness and potential for bioactive functionalisation. Commonly, platinum(IV) complexes are prepared by oxidation of the platinum(II) precursors with hydrogen peroxide [46,47]. This synthetic step generates the platinum(IV) scaffold, incorporating two axial positions with two hydroxides coordinated and readily available for substitution by bioactive ligands. While cisplatin, carboplatin and oxaliplatin are being exploited as precursor cores for new platinum(IV) complexes [48,49,50,51], we contribute to this approach by utilising the potent platinum(II) complexes, **PHEN*SS***, **5ME*SS*** and **56ME*SS***. With the vast number of bioactive ligands available to coordinate to platinum(IV), the indole-based derivative, 5-benzyloxyindole-3-acetic acid (5B3A) caught our interest. We have selected 5B3A as axial ligand because it incorporates an indole pharmacophore, which has been used in drug design and discovery [52,53]. The bicyclic nature of the compound exhibits multiple therapeutic properties [54,55]; and for this reason, it is found in many drugs, specifically those that target cancer [56]. Recent studies have also shown the effective use of the indole scaffold in creating new metal-based anticancer drugs with improved activity [57,58]. We expect that by coordinating 5B3A to the platinum(IV) form of **PHEN*SS***, **5ME*SS*** and **56ME*SS***, we will enhance the anticancer properties by providing an additional mode of action against cancer cells.

Herewith, we report the syntheses and structural characterisation of three platinum(IV) complexes of type **[Pt^IV^(H_L_)(A_L_)(X)(OH)]^2+^**, where **X** is the indole-based axial ligand, 5B3A: [Pt^IV^(phen)(*SS*-DACH)(5B3A)(OH)]^2+^ (**[PHEN*SS*(IV)(5B3A)(OH)]^2+^** = **P-5B3A**), [Pt^IV^(5-Mephen)(*SS*-DACH)(5B3A)(OH)]^2+^ (**[5ME*SS*(IV)(5B3A)(OH)]^2+^** = **5-5B3A**), and [Pt^IV^(5,6-Me_2_phen)(*SS*-DACH)(5B3A)(OH)]^2+^ (**[56ME*SS*(IV)(5B3A)(OH)]^2+^** = **56-5B3A**) (Figure 4). The purity and chemical structures of the complexes were confirmed through high-performance liquid chromatography (HPLC), nuclear magnetic resonance (^1^H-NMR; two-dimensional correlation spectroscopy (2D-COSY); heteronuclear multiple quantum correlation (^1^H-^195^Pt-HMQC)), ultraviolet–visible (UV), circular dichroism (CD) and elemental microanalysis experiments. All complexes were purified via a flash chromatography system. To determine the anticancer activity of the resultant complexes, they were screened against twelve human cell lines including HT29 colon, U87 glioblastoma, MCF-7 breast, A2780 ovarian, H460 lung, A431 skin, Du145 prostate, BE2-C neuroblastoma, SJ-G2 glioblastoma, MIA pancreas, the cisplatin-resistant ADDP ovarian variant, and the non-tumour-derived MCF10A breast line. In addition to solubility, stability and lipophilicity studies, the reduction behaviour of **P-5B3A**, **5-5B3A** and **56-5B3A** upon treatment with ascorbic acid (AsA) was also determined with the aid of ^1^H-NMR and one-dimensional ^195^Pt-NMR (1D-^195^Pt-NMR). The capacity of the complexes to generate reactive oxygen species (ROS), alter mitochondrial activity via causing mitochondrial membrane potential (MtMP) changes, and inhibit histone deacetylase (HDAC) were also investigated to determine how these parameters may affect their overall biological activity.

## 2. Results and Discussion

### 2.1. Syntheses and Characterisation

The platinum(II) precursors and platinum(IV) scaffolds of **[Pt^II^(H_L_)(A_L_)]^2+^** and **[Pt^IV^(H_L_)(A_L_)(OH)_2_]^2+^** were prepared as described in the literature without modifications [59]. The NHS ester of 5B3A was also prepared as described in the literature [60]; and to confirm its synthesis, HPLC, ^1^H-NMR and 2D-COSY experiments were undertaken (Appendix A). The ^1^H-NMR and 2D-COSY spectra of the uncoordinated axial ligand, 5B3A were also acquired for reference and used to assist in assigning the protons for its NHS ester derivative, NHS-5B3A (Appendix A). To synthesise the proposed platinum(IV) complexes, **[Pt^IV^(H_L_)(A_L_)(X)(OH)]^2+^** (**P-5B3A**, **5-5B3A** and **56-5B3A**) (Figure 5), established protocols were adapted [61]. The resultant complexes were purified via a flash chromatography system to acquire purity of above 95%. The purity and chemical structures of **P-5B3A**, **5-5B3A** and **56-5B3A** were confirmed by HPLC (Appendix A), ^1^H-NMR (Appendix A), 2D-COSY (Appendix A), ^1^H-^195^Pt-HMQC (Appendix A), UV (Appendix A), CD (Appendix A) and elemental microanalysis experiments.

#### 2.1.1. NMR Assignments

NMR techniques including ^1^H-NMR, 2D-COSY and ^1^H-^195^Pt-HMQC were applied to confirm the chemical structures of the synthesised platinum(IV) complexes, **P-5B3A**, **5-5B3A** and **56-5B3A** (Appendix A). The ^1^H-NMR and ^1^H-^195^Pt-HMQC assignments of **P-5B3A**, **5-5B3A** and **56-5B3A** are summarised in the Materials and Methods section. Because DMSO-d_6_ was used for the NMR experiments, amine proton resonances were observed in the aromatic region (8–10 ppm) (Appendix A). The acquired NMR data are comparable to published data [40,61,62,63].

As depicted in the ^1^H-NMR spectrum of **P-5B3A** (Appendix A), the protons of the heterocyclic ligand, phen, resonated between 8 and 10 ppm. The most deshielded or downfield phen protons, H2 and H9, were assigned to the doublets at 9.40 and 9.38 ppm, which exhibit an equal *J*-coupling constant of 5.4 Hz. The doublet of doublets at 9.06 ppm was assigned to H4 and H7 protons, and both have *J*-coupling constants of 4.8 and 8.1 Hz, respectively. Subsequently, a multiplet was recorded at 8.32 ppm which was induced by H3 and H8 protons. The remaining protons in the phen ligand, H5 and H6, were assigned to the sharp singlet at 8.26 ppm. It is also noteworthy that the broad signals adjacent to the signals of H4, H7, H3, H8, H2 and H9 protons were caused by the amines of the chiral ancillary ligand, *SS*-DACH (Appendix A).

When comparing the multiplicities demonstrated for **P-5B3A** in the aromatic region with the multiplicities for its corresponding platinum(II) precursor and platinum(IV) scaffold, **PHEN*SS*** and **PHEN*SS*(IV)(OH)_2_**, slight variations are evident as shown in Figure 6. Upon coordinating 5B3A to the platinum(IV) scaffold, **PHEN*SS*(IV)(OH)_2_**, an upfield shift was detected for the following phen protons: H4, H7, H3, H8, H5 and H6 (Figure 6). This phenomenon may have been caused by the possible π–π stacking interactions between the aromatic rings of 5B3A and the phen, and this upfield movement was also observed by structurally related complexes in the literature [59,61]. Overall, the variations in multiplicity reported for **P-5B3A** is primarily influenced by the coordination of the axial ligand, 5B3A.

Notable differences in chemical multiplicity were also demonstrated by the protons of 5B3A originating from the benzene and indole bicyclic rings (benzene fused with pyrrole) (Figure 6 and Appendix A). The pyrrole proton (H_pyrrole_) proved to be the most deshielded as it resonated furthest downfield as a singlet at 10.3 ppm, as shown in Figure 6 and Appendix A. The distorted multiplicities between 7.37–7.53 ppm were assigned to the benzene protons, a, b, c, d and e (Figure 6 and Appendix A). The irregular doublet and doublet of doublets at 7.02 and 6.69 ppm were assigned to the indole protons, i and h, respectively. As for the remaining indole protons, f and g, a singlet at 6.46 ppm was recorded, which indicates overlapping. Of further note, the 2D-COSY of **P-5B3A** was also analysed to elucidate the assignment of the indole protons of 5B3A (Appendix A). As shown in Appendix A, f and h proved to be the only protons coupled to H_pyrrole_. Proton h was found to be coupled to protons g and i (Appendix A). Furthermore, the methylene protons of 5B3A, β and α, was assigned to the singlet and the multiplet at 4.91 and 3.23 ppm, respectively (Appendix A). Within the aliphatic region (1–3 ppm), the recorded resonances exhibited by the protons of the *SS*-DACH (Appendix A) are also consistent with literature data [40,63,64].

^1^H-^195^Pt-HMQC experiments were also conducted for **P-5B3A** at 400 and −2800 ppm (Appendix A), to further validate the coordination of 5B3A to **PHEN*SS*(IV)(OH)_2_**. The platinum(II) precursors, **[Pt^II^(H_L_)(A_L_)]^2+^** typically resonate at −2800 ppm while their equivalent platinum(IV) scaffolds, **[Pt^IV^(H_L_)(A_L_)(OH)_2_]^2+^** resonate within 400 ppm [40,59,63]. The ^1^H-^195^Pt-HMQC spectrum of **P-5B3A** highlights the correlation of protons H2 and H9 (9.12 ppm) and of H3 and H8 (8.10 ppm) with the platinum peaks at 498 ppm (Appendix A). No peak was found at −2800 ppm (Appendix A), which confirmed the absence of platinum(II) species. This further supports that not only the sample used for the NMR experiments are free of platinum(II) species denoting purity, the 5B3A ligand was also successfully coordinated to **PHEN*SS*(IV)(OH)_2_** and occupied one axial position.

The ^1^H-NMR assignment of multiplicities for the spectra of **5-5B3A** and **56-5B3A** (Appendix A) followed the same rationale to that discussed for **P-5B3A**. With respect to the ^1^H-NMR spectra of **5-5B3A** and **56-5B3A** (Appendix A), some distinctions in multiplicity were evident when compared to those observed for **P-5B3A**, most specifically in the aromatic region. While these distinctions are mostly attributed to the coordination 5B3A ligand, another factor that contributes to these is the methylation of the heterocyclic rings of **5-5B3A** and **56-5B3A**. This phenomenon has been observed previously with similar complexes in the literature [59,61,65]. Finally, the ^1^H-^195^Pt-HMQC spectra of **5-5B3A** and **56-5B3A** (Appendix A) also confirmed the successful coordination of 5B3A to the respective platinum(IV) scaffolds, indicating the axial ligand only occupying one axial position.

In addition to NMR, elemental microanalysis was performed on an Elementar Vario MICRO (Elementar Analysensysteme GmbH) to further confirm the chemical composition and purity of **P-5B3A**, **5-5B3A** and **56-5B3A**. The resulting elemental microanalysis is consistent with the calculated values for the prodrugs as nitrates with co-crystallised H_2_O molecules, as summarised in the Materials and Methods section for **P-5B3A.2H_2_O**, **5-5B3A.3H_2_O** and **56-5B3A.2H_2_O**.

#### 2.1.2. UV and CD Spectral Analysis

The molecular electronic transitions exhibited by **P-5B3A**, **5-5B3A** and **56-5B3A** in the UV measurements are comparable to published data (Appendix A) [40,61,62,66]. The acquisition of UV spectra was achieved as previously described [59,61,65,67]. Experiments were obtained in triplicates; three separate stock solutions were prepared for each complex. All spectra were baseline corrected. Mean extinction coefficients (ε) were determined with standard deviation and errors based on the constructed plot curves. CD measurements were also undertaken to confirm that the chirality of the ancillary ligand, *SS*-DACH, was retained by **P-5B3A**, **5-5B3A** and **56-5B3A** during syntheses. Furthermore, the absorption bands exhibited in the CD spectra of **P-5B3A**, **5-5B3A** and **56-5B3A** (Appendix A) are also comparable to published data [40,61,62,66]. Due to the structural similarities of the complexes, no significant differences were observed in their UV or CD spectra. All characteristic peaks exhibited by **P-5B3A**, **5-5B3A** and **56-5B3A** in their UV and CD spectra are summarised in Table 1.

The anticipated metal-to-ligand charge transfer interactions and π–π* transitions were exhibited in the UV spectra of the complexes (Appendix A), and are consistent with similar complexes in the literature [40,61,62,66]. These interactions are the effect of the heterocyclic ligands of the complexes, phen, 5-Mephen and 5,6-Me_2_phen which exhibit ligand-centred π–π* transitions. The patterns observed in the UV spectra of **P-5B3A**, **5-5B3A** and **56-5B3A** (Figure 7 and Appendix A), especially the bathochromic shift, are heavily influenced by their heterocyclic ligands. This observation corresponds to published data [40,61,62,66]. Prominent peaks were detected at the wavelengths ~200 and ~279–289 nm for all complexes, as shown in Figure 7.

Based on the UV spectra of **P-5B3A** (Figure 7 and Appendix A), a prominent absorption band was detected at 204 nm, which was followed by a dip between ~240 and 250 nm, denoting a hypochromic shift. Subsequently, a hyperchromic shift was observed when a less prominent peak appeared at 279 nm, which was followed by a weak band between 300–310 nm, as shown in Figure 7. The absorption bands detected for **P-5B3A** were also evident in the UV spectra of **5-5B3A** and **56-5B3A** (Figure 7, Appendix A). Moreover, the absorptions bands shifted towards a longer wavelength, which indicates bathochromic shift. Interestingly, a characteristic band was observed for **56-5B3A** between ~240 and 250 nm which appears like a weak shoulder. Finally, it is also noteworthy that the characteristic absorption bands for uncoordinated 5B3A are evident within the absorption bands for the complexes (Figure 7). This provides further evidence of the contributions of 5B3A when coordinated to platinum. Overall, the UV spectra demonstrated by **P-5B3A**, **5-5B3A**, **56-5B3A** also show evidence of the presence or lack of methyl groups on the heterocyclic ligands.

CD measurements were also performed to confirm that the chirality of the chiral ancillary ligand, *SS*-DACH was retained during syntheses. Chirality is an essential feature that directly contributes to the potency of the studied complexes, and this has been the subject of previous studies [33,34,35,68]. The CD spectra of **P-5B3A**, **5-5B3A** and **56-5B3A** (Appendix A), demonstrated no significant differences. Strong peaks were evident at lower wavelengths, which is consistent with previously reported structurally similar platinum(IV) complexes [40,61,62,66]. The CD spectra of **P-5B3A** (Appendix A), exhibited prominent negative absorption bands (200–240 nm), followed by weaker positive absorption bands between 250 and 280 nm. This pattern was also observed in the CD spectra of **5-5B3A** (Appendix A) and **56-5B3A** (Appendix A), although **56-5B3A** exhibited slightly more prominent positive absorption bands between 250 and 280 nm.

### 2.2. Physicochemical Investigations

#### 2.2.1. Lipophilicity

The order of complexes by increasing lipophilicity is **P-5B3A** < **5-5B3A** < **56-5B3A** (Table 2 and Appendix A). It can be deduced that increasing methylation on the heterocyclic ligand increased lipophilicity. This trend agrees with published data [59,61,65,66,69]. The determined log k_w_ values are consistent and within the range of the log k_w_ values determined for similar complexes reported in the literature [59,61]. Furthermore, cisplatin has a reported log k_w_ value of −2.19 [70,71], which is significantly lower than the acquired log k_w_ values of **P-5B3A**, **5-5B3A** and **56-5B3A** (Table 2). This suggests that **P-5B3A**, **5-5B3A** and **56-5B3A** are expected to have better absorption properties than cisplatin.

#### 2.2.2. Preliminary Reduction Measurements

^1^H-NMR and 1D-^195^Pt-NMR spectroscopy were utilized to investigate the reduction activity of the platinum(IV) complexes, **P-5B3A**, **5-5B3A** and **56-5B3A**, using previously reported methods [59,61,65]. The biological reductant selected for this study is AsA. Briefly, sequential ^1^H-NMR experiments were performed for 1 h at 37 °C, proceeded by 1D-^195^Pt-NMR at 400 and −2800 ppm (30 min per region). It is noteworthy that the purity of each complex was examined prior the reduction experiments ensuring that no traces of platinum(II) species were evident in the samples used. All complexes were dissolved in PBS and DMSO-d_6_, without the reductant. Preliminary 1D-^195^Pt-NMR measurements were conducted at 37 °C at 400 and −2800 ppm. This was completed to ensure that the PBS and DMSO-d_6_ do not interfere with the complexes by causing unwanted reduction. From the preliminary 1D-^195^Pt-NMR spectra, it was confirmed that the PBS and DMSO-d_6_ did not cause the complexes to reduce (Appendix A). Upon treatment of the platinum(IV) complexes with AsA, there was considerable reduction observed in the ^1^H-NMR and 1D-^195^Pt-NMR spectra acquired (Appendix A). Notable changes in chemical multiplicity were recorded in the ^1^H-NMR reduction spectra of **P-5B3A**, **5-5B3A** and **56-5B3A** (Appendix A), especially along the aromatic region. The estimated reduction time points for **P-5B3A**, **5-5B3A** and **56-5B3A** are also outlined in Table 3.

The expanded ^1^H-NMR reduction spectra for **P-5B3A** at different time intervals showing distinct changes in chemical multiplicity are highlighted in Figure 8. The separate doublets exhibited by H2 and H9 protons and the multiplet exhibited by H4 and H7 protons (in the platinum(IV) form) resolved into a doublet (Figure 8). Subsequently, an upfield shift was observed for H3, H8, H5 and H6 protons. Notably, the H_pyrrole_ of the axial ligand, 5B3A appeared further downfield at 10.8 ppm upon dissociating from platinum. Another notable change in multiplicity was demonstrated by the indole protons, f and g. As shown in Figure 8, these protons resonated as a singlet at 6.46 ppm (in the platinum(IV) form), but eventually resolved into two separate singlets as it dissociated from the platinum. These changes in multiplicity show that the platinum(IV) complex, **P-5B3A** successfully reduced to its core platinum(II) precursor, **PHEN*SS***. To further confirm this, 1D-^195^Pt-NMR measurements were performed at 37 °C at 400 and −2800 ppm immediately after the final ^1^H-NMR measurement. At approximately 1.5 h, the 1D-^195^Pt-NMR reduction spectra of **P-5B3A** was acquired and no platinum(IV) peak was found at 400 ppm, as shown in Appendix A. However, at −2800 ppm, 30 min after the first 1D-^195^Pt-NMR reduction spectra were acquired, a prominent peak was recorded, which indicates the presence of the platinum(II) precursor, **PHEN*SS*** (Appendix A). The changes in multiplicity observed in the ^1^H-NMR reduction spectra of **P-5B3A** (Figure 8 and Appendix A) were comparable to those observed for **5-5B3A** and **56-5B3A** (Appendix A). **5-5B3A** and **56-5B3A** were also confirmed to have successfully reduced to their core platinum(II) precursors, **5ME*SS*** and **56ME*SS***, respectively, based on the acquired 1D-^195^Pt-NMR reduction spectra (Appendix A). Because **P-5B3A**, **5-5B3A** and **56-5B3A** are reduced to their core platinum(II) precursors, this further confirms that they are prodrugs.

With respect to the estimated reduction time points in Table 3, there was a notable difference in the rate of reduction exhibited by the complexes (Table 3). **P-5B3A** was the only complex to have reduced more slowly, with 50% reduction occurring between 10 and 15 min and maximal reduction at 60 min (Figure 8 and Appendix A). In comparison, **5-5B3A** reduced by 50% between 5 and 10 min, while maximal reduction occurred at 30 min (Appendix A). Interestingly, **56-5B3A** was confirmed to have the fastest rate of reduction with maximal reduction at 10 min (Appendix A). This indicates that the differences in reduction rates are primarily regulated by the heterocyclic ligands rather than the coordinated 5B3A ligand. This trend has also been observed previously with similar complexes in the literature [59,61]. These preliminary results provided insight into how these complexes reduce in the presence of a reducing agent like AsA. We expect the complexes to reduce differently under physiological conditions (i.e., in the presence of human serum or plasma).

#### 2.2.3. Stability Measurements

The stability of **P-5B3A**, **5-5B3A** and **56-5B3A** was monitored via analytical HPLC. Surprisingly, **P-5B3A** and **5-5B3A** did not show any signs of significant hydrolysis for up to 1 week according to the HPLC chromatograms obtained (Appendix A), only minimal peak traces at ~5 min denoting the presence of the equivalent platinum(II) precursors. While **56-5B3A** did not completely hydrolyse in solution, a trace platinum(II) peak was also recorded at ~6 min which gradually became more prominent over the week (Appendix A). Overall, the platinum(IV) complexes, **P-5B3A**, **5-5B3A** and **56-5B3A** proved to be adequately stable in aqueous solution for up to 1 week at 37 °C without any significant hydrolysis involved.

### 2.3. Biological Investigations

#### 2.3.1. In Vitro Anticancer Activity

The in vitro anticancer activity of the synthesised platinum(IV) complexes, **P-5B3A**, **5-5B3A** and **56-5B3A**, including the uncoordinated axial ligand, 5B3A was determined in multiple human cell lines such as HT29 colon, U87 glioblastoma, MCF-7 breast, A2780 ovarian, H460 lung, A431 skin, Du145 prostate, BE2-C neuroblastoma, SJ-G2 glioblastoma, MIA pancreas, the cisplatin-resistant ADDP ovarian variant, and the non-tumour-derived MCF10A breast line using the MTT assay [59,61,65,72]. The cisplatin-resistant ADDP ovarian variant was included for this study to evaluate whether the complexes can circumvent resistance. All cytotoxicity values are expressed as GI_50_. The cytotoxicity values for 5B3A, **P-5B3A**, **5-5B3A** and **56-5B3A**, together with cisplatin, oxaliplatin and carboplatin, are summarised in Table 4. The cytotoxicity values of cisplatin, oxaliplatin and carboplatin were acquired from our previous studies [59,61,65], and have been included for comparison purposes. Additionally, the cytotoxicity values for platinum(II) precursors (**PHEN*SS***, **5ME*SS*** and **56ME*SS***) and platinum(IV) scaffolds (**PHEN*SS*(IV)(OH)_2_**, **5ME*SS*(IV)(OH)_2_** and **56ME*SS*(IV)(OH)_2_**) are also summarised in Table 5 for reference [59,61,65].

**P-5B3A**, **5-5B3A** and **56-5B3A** proved to be potent in most cell lines (Table 4), although **P-5B3A** was least active in the BE2-C neuroblastoma cell line, with a GI_50_ value of 13,000 ± 5000 nM. **5-5B3A** and **56-5B3A** were found to be more active than cisplatin, oxaliplatin and carboplatin in all the cell lines, as shown in Table 4. The average GI_50_ values of the complexes were calculated from all cell lines and with the order of increasing cytotoxicity: **P-5B3A** < **5-5B3A** < **56-5B3A**. The complexes incorporating the methylated heterocyclic ligands, **5-5B3A** and **56-5B3A** demonstrated greater anticancer potential compared to **P-5B3A**. This trend in cytotoxicity was anticipated, as this has been previously observed with the equivalent platinum(II) precursors (**PHEN*SS***, **5ME*SS*** and **56ME*SS***) and platinum(IV) scaffolds (**PHEN*SS*(IV)(OH)_2_**, **5ME*SS*(IV)(OH)_2_** and **56ME*SS*(IV)(OH)_2_**) (Table 5), as well as other similar complexes in the literature [31,38,59,61,62,65,66,69,73]. These results corroborate the notion that increasing methylation of the heterocyclic ligand enhances the biological activity of the complexes.

**P-5B3A** exhibited an average GI_50_ value of 2084 ± 703 nM, which is 20- and 50-fold larger than the average GI_50_ values calculated for its derivatives, **5-5B3A** (104 ± 12 nM) and **56-5B3A** (42 ± 7 nM), respectively (Table 4). **P-5B3A** was more potent than cisplatin, oxaliplatin and carboplatin in most cell lines; for example, **P-5B3A** was at least 57-, 5- and 250-fold more potent than cisplatin (11,300 ± 1900 nM) (** *p* < 0.01), oxaliplatin (900 ± 200 nM) and carboplatin (>50,000 nM), respectively, in the HT29 colon cell line (Table 4). In the cisplatin-resistant ADDP ovarian variant cell line, **P-5B3A** (330 ± 8.8 nM) was at least 85-fold more potent than cisplatin (28,000 ± 1600 nM) (**** *p* < 0.0001), and 152-fold more potent than carboplatin (>50,000 nM) (Table 4). **5-5B3A** also demonstrated potency in most cell lines, most notably in the Du145 prostate, HT29 colon and ADDP. In the Du145 prostate cell line, **5-5B3A** was 120-, 290- and 1500-fold more potent than cisplatin (1200 ± 100 nM) (** *p* < 0.01), oxaliplatin (2900 ± 400 nM) and carboplatin (15,000 ± 1200 nM), respectively (Table 4). Moreover, **5-5B3A** elicited a GI_50_ value of 18 ± 0.67 nM in the HT29 colon cell line as shown in Table 4, which makes cisplatin almost 630-fold less effective in this cell line (** *p* < 0.01). Of further note, **5-5B3A** was also found to be almost 1220-fold more effective at inhibiting ADDP ovarian variant cells compared to cisplatin (**** *p* < 0.0001) (Table 4). **56-5B3A** proved to be the most biologically active platinum(IV) derivative, its potency was significant in all cell lines with GI_50_ values ranging between 1.2 and 150 nM (Table 4). Remarkably, **56-5B3A** (1.2 ± 0.6 nM) was 1000-fold more potent than cisplatin (** *p* < 0.01), and at least 2400- and 12,500-fold more potent than oxaliplatin and carboplatin, respectively, in the Du145 prostate cell line. In addition, **56-5B3A** also produced impressive cytotoxicity on the following cell lines: HT29 colon (4.4 ± 1.6 nM), H460 lung (7.8 ± 2.2 nM) and MIA pancreas (12 ± 3 nM), as shown in Table 4.

The coordination of 5B3A in the axial position of the platinum(IV) scaffolds (**PHEN*SS*(IV)(OH)_2_**, **5ME*SS*(IV)(OH)_2_** and **56ME*SS*(IV)(OH)_2_**) resulted in enhanced overall biological activity of the complexes (**P-5B3A**, **5-5B3A** and **56-5B3A**). For example, the average GI_50_ value of **PHEN*SS*(IV)(OH)_2_** in all cell lines was 3318 ± 880 nM (Table 5) and this decreased by almost 1.6 upon coordinating 5B3A to form the platinum(IV) derivative, **P-5B3A** (2084 ± 703 nM). The average GI_50_ values of **5ME*SS*(IV)(OH)_2_** and **56ME*SS*(IV)(OH)_2_** also decreased by 4 and 3.6 fold when compared to the GI_50_ values of their platinum(IV) derivatives, **5-5B3A** and **56-5B3A,** respectively (Table 4 and Table 5). Furthermore, the cytotoxicity of the platinum(II) precursors (**PHEN*SS***, **5ME*SS*** and **56ME*SS***) and their platinum(IV) derivatives (**P-5B3A**, **5-5B3A** and **56-5B3A**) in most cell lines are comparable. Intriguingly, **P-5B3A** was less potent than **PHEN*SS*** in all cancer cell lines (Table 4). In the MCF10A normal breast cell line, only a 0.9-fold difference in cytotoxicity was reported between **PHEN*SS*** (300 ± 58 nM) and **P-5B3A** (280 ± 39 nM) (Table 4 and Table 5). Whereas **5-5B3A** and **56-5B3A** were found to be marginally more potent than their platinum(II) precursors, **5ME*SS*** and **56ME*SS,*** respectively, in most cell lines. **5-5B3A** has an average GI_50_ value of 104 ± 12 nM, which is approximately 0.9-fold lower than the average GI_50_ value determined for **5ME*SS*** (117 ± 12 nM). Lastly, **56-5B3A** was confirmed to be slightly more effective than **56ME*SS*** in inhibiting HT29 colon, Du145 prostate and ADDP cell populations (Table 4 and Table 5). **56-5B3A** was 0.4-, 0.3- and 0.5-fold more potent than **56ME*SS*** in these cell lines.

The selectivity cytotoxicity index (SCI) of the synthesised platinum(IV) complexes, **P-5B3A**, **5-5B3A** and **56-5B3A**, and cisplatin was also calculated by dividing their GI_50_ in the normal breast cancer cell line (MCF10A) to their GI_50_ in all cancer cell lines (Table 6). Generally, a higher SCI corresponds to improved selectivity towards cancer cells [74,75]. The complexes demonstrated a degree of selectivity towards the HT29 colon, Du145 prostate and the cisplatin-resistant ADDP cell lines relative to cisplatin (Table 6). This also corresponds to the low GI_50_ values elicited by the complexes in these cell lines (Table 4). In the HT29 colon cell line, **P-5B3A** and **5-5B3A** had a calculated SCI of 1.40 and 1.06, respectively, while for **56-5B3A**, the calculated SCI was 0.75. In the same cell line, cisplatin was less selective with a calculated SCI of 0.46. Moreover, while it was expected that cisplatin had a lower SCI (0.19) in the cisplatin-resistant ADDP cell line, the SCI of **P-5B3A**, **5-5B3A** and **56-5B3A** ranged between 0.55 and 0.85 (Table 6). Interestingly, cisplatin had a higher SCI (4.33) compared to the complexes (1.47–2.75) in the Du145 prostate cell line, as shown in Table 6. What is also noteworthy is that cisplatin had higher SCI in most of the reported cell lines (Table 6).

Despite the impressive in vitro cytotoxicity exhibited by **P-5B3A**, **5-5B3A** and **56-5B3A** in most cell lines, it is important to note that these complexes are not selective (Table 6). For example, in the MCF10A normal breast cell line, the complexes were significantly more toxic than in the MCF-7 cancerous breast cell line (Table 4). Additionally, the prodrugs, and their corresponding platinum(IV) scaffolds and platinum(II) precursors, exhibited decreased cytotoxicity in the BE2-C neuroblastoma and SJ-G2 glioblastoma cell lines compared to the other cell lines tested (Table 4 and Table 5). This decreased cytotoxicity may be associated with efflux resistance mechanisms by the cell lines. P-glycoproteins are multidrug resistance proteins that are upregulated in the BE2-C neuroblastoma and SJ-G2 glioblastoma cell types [76], and these may prevent the adequate accumulation of the complexes in these cell lines. Nevertheless, the complexes demonstrated a degree of selectivity towards the HT29 colon, Du145 prostate and the cisplatin-resistant ADDP ovarian variant cell lines, which has been a consistent trend observed with similar complexes in the literature [59,61,65,66,69]. Their capacity to overcome resistance was confirmed based on their responsiveness to the cisplatin-resistant ADDP ovarian variant cell line. The coordination of 5B3A to the platinum(IV) cores of **PHEN*SS***, **5ME*SS*** and **56ME*SS*** enhanced the biological activity of the resultant complexes (**P-5B3A**, **5-5B3A** and **56-5B3A**) in most cell lines tested (Table 4 and Table 5).

#### 2.3.2. Drug-Induced ROS Production as a Treatment Strategy

ROS embody a variety of chemically reactive free and non-free radicals which are by-products of aerobic metabolism [77]. These metabolic intermediates are mostly generated in the mitochondria [78,79]. ROS are highly versatile for having important functions in cell signaling and other physiological cascades that contribute to maintaining cellular life [80,81]. Additionally, ROS levels are tightly regulated and constantly maintained at basal levels in the normal cell environment [82,83]. Regulation is critical to prevent excessive release of ROS from inducing oxidative stress, causing damage to cell membranes, proteins, ribonucleic acid, and DNA [84,85]. Other sources that can also elevate levels of ROS are exogenous substances such as drugs, xenobiotics, and radiation pollutants [85].

Elevated ROS levels are disadvantageous for normal healthy cells because oxidative stress would transpire. In contrast, this is advantageous for cancer cells to an extent as cancer cells are equipped with greater levels of antioxidant enzymes to neutralise excessive amount of ROS during proliferation [86]. Elevated levels of ROS can promote cancer development and progression. However, cancer cells also become vulnerable to different death-inducing pathways when there is a disproportional increase in ROS levels especially when exposed for prolonged periods [87]. This can be achieved by introducing anticancer drugs. Most chemotherapeutics can increase ROS intracellularly, and this treatment approach has the potential to enhance cancer-killing effects [87,88,89,90]. We hypothesised that **P-5B3A**, **5-5B3A** and **56-5B3A** would produce higher levels of ROS than their equivalent platinum(II) precursors (**PHEN*SS***, **5ME*SS*** and **56ME*SS***) and platinum(IV) scaffolds (**PHEN*SS*(IV)(OH)_2_**, **5ME*SS*(IV)(OH)_2_** and **56ME*SS*(IV)(OH)_2_**) because of the coordinated 5B3A axial ligand. The indole core of 5B3A has garnered synthetic interest because of the numerous pharmacological potencies it offers [52,53,55], and one of them is the generation of ROS [91,92].

The ROS generation capacity of **P-5B3A**, **5-5B3A** and **56-5B3A**, 5B3A, and cisplatin was determined in HT29 colon cells at different time intervals: 24, 48 and 72 h (Table 7 and Figure 9, Appendix A). All values are expressed as relative fluorescence units (RFU). With reference to the control, a greater RFU corresponds to elevated ROS production (increased fluorescence of DCF). Previously published RFU values of the platinum(II) precursors (**PHEN*SS***, **5ME*SS*** and **56ME*SS***) and platinum(IV) scaffolds (**PHEN*SS*(IV)(OH)_2_**, **5ME*SS*(IV)(OH)_2_** and **56ME*SS*(IV)(OH)_2_**) were also included in Table 7 and Figure 9, for reference [59,61,65]. The HT29 colon cell line was nominated for testing because **P-5B3A**, **5-5B3A** and **56-5B3A** displayed a degree of selectivity in this cell line (Table 4 and Table 6).

Relative to the control, the synthesised platinum(IV) complexes, **P-5B3A**, **5-5B3A** and **56-5B3A** caused a notable increase in ROS levels by 24 h, and progressively continued until completion of the experiments at 72 h in the HT29 colon cell line (Table 7 and Figure 9 and Appendix A). **P-5B3A** and **5-5B3A** displayed similar RFU values at 24 and 48 h. At 24 h, **P-5B3A** incurred a slightly lower RFU value (156) compared to **5-5B3A** (160) (Table 7). Subsequently, at 48 h, the RFU value reported for **P-5B3A** increased to 214, which is slightly lower than the RFU value reported for **5-5B3A** (230), as shown in Table 7. At 72 h, the RFU values reported for **P-5B3A** and **5-5B3A** considerably increased to 315 and 474, respectively. The most potent platinum(IV) derivative, **56-5B3A** caused the greatest ROS production compared to **P-5B3A** and **5-5B3A** (Table 7 and Figure 9, Appendix A (compared to cisplatin)). At 24 and 48 h, **56-5B3A** elicited RFU values of 274 and 418, respectively, which are greater than the RFU values reported for **P-5B3A** and **5-5B3A** in that time period, as shown in Table 7. Peak ROS production was demonstrated at 72 h for **56-5B3A**, with an RFU value of 474 (Table 7). The 5B3A ligand alone generated ROS (RFU values of 138, 153 and 164) (Table 7 and Figure 9, Appendix A), but to a lesser extent when compared to **P-5B3A**, **5-5B3A** and **56-5B3A**. It is also noteworthy that there may be a link between ROS production at 72 h (Table 7 and Figure 9, Appendix A) and in vitro cytotoxicity, in the HT29 colon cell line (Table 4). At 72 h, the least potent platinum(IV) derivative (**P-5B3A**) incurred an RFU value of 246, while the more potent platinum(IV) derivative (**5-5B3A**) incurred an RFU value of 315 (Table 7). Moreover, the most potent platinum(IV) derivative (**56-5B3A**) caused the greatest ROS production at 72 h with a reported RFU value of 474, as shown in Table 7. This trend was also observed with similar complexes in the literature [59,65]. It is noteworthy that the GI_50_ values for each complex were used for the ROS experiments, so the same concentration of each complex is not directly compared. With the exception of **56-5B3A**, the reduced effect of the prodrugs, **P-5B3A** and **5-5B3A**, as well as the platinum(II) precursors at 72 h may have been due to their cytostatic effects. Overall, the findings indicate that **P-5B3A**, **5-5B3A** and **56-5B3A** are effective at causing elevated levels of ROS for up to 72 h. While there were no significant differences observed between these platinum(IV) complexes cisplatin in terms of ROS production (Appendix A), the increase in ROS from 24, 48 to 72 h for **56-5B3A** was sill greater than that of cisplatin. Nevertheless, through this mechanism, the reported platinum(IV) complexes are still expected to trigger oxidative stress inside cancer cells, hence effectively ceasing cancer development. Our findings are also comparable to the results obtained in a similar study undertaken by Tolan et al. [92], where they have demonstrated the coordination of a ROS-inducing indole-based ligand to the core of a platinum complex (cisplatin), produced prodrugs with enhanced anticancer killing effects via the effective production of ROS.

#### 2.3.3. Drug-Induced Mitochondrial Dysfunction as a Treatment Strategy

The mitochondria are a key feature of in cell types and are primarily responsible for producing energy required for metabolism [93,94]. While it is often described as a powerhouse, the mitochondria are also susceptible to oxidative stress [95,96]. Several drug classes, including chemotherapy drugs have been identified to promote disturbance in mitochondrial functions [97,98]. When this occurs, the mitochondrial membrane undergoes permeabilisation, which in turn creates an imbalance in MtMP [99,100]. MtMP is a parameter that reflects mitochondrial homeostasis, clearly because it is involved in the energy storage process during oxidative phosphorylation [101,102]. Fluctuations in MtMP are a consequence of normal physiological activity; however, prolonged fluctuations, either a significant increase or decrease in MtMP, prove to be deleterious to the mitochondria as this effect can trigger the release of apoptotic factors that lead to cell death [103,104,105].

Because the majority of ROS is generated in the mitochondria [78,106], it was appropriate to assess any mitochondrial activity changes upon treatment with the synthesised platinum(IV) complexes, **P-5B3A**, **5-5B3A** and **56-5B3A**. This was achieved by determining if the overproduction of ROS induced by these complexes impacts mitochondrial activity by altering MtMP. Briefly, there is evidence showing a strong correlation between increased levels of ROS and mitochondrial impairment [106,107,108,109,110]. We determined MtMP changes induced by **P-5B3A**, **5-5B3A** and **56-5B3A**, 5B3A, the platinum(II) precursors (**PHEN*SS***, **5ME*SS*** and **56ME*SS***), the platinum(IV) scaffolds (**PHEN*SS*(IV)(OH)_2_**, **5ME*SS*(IV)(OH)_2_** and **56ME*SS*(IV)(OH)_2_**), and cisplatin in HT29 colon cells at different time intervals: 24, 48 and 72 h (Table 8 and Figure 10 and Appendix A). The MtMP changes recorded are measured as RFU. Relative to the control, a lower or greater RFU is indicative of mitochondrial dysfunction caused by unstable MtMP.

In summary, the platinum(IV) complexes, **P-5B3A**, **5-5B3A**, **56-5B3A** and the 5B3A ligand, together with their equivalent platinum(II) precursors (**PHEN*SS***, **5ME*SS*** and **56ME*SS***) and platinum(IV) scaffolds (**PHEN*SS*(IV)(OH)_2_**, **5ME*SS*(IV)(OH)_2_** and **56ME*SS*(IV)(OH)_2_**), and cisplatin caused a progressive decline in mitochondrial activity for up to 72 h, as reflected by the incurred RFU values (Table 8 and Figure 10 and Appendix A). Interestingly, the pattern observed here may be linked to the elevated ROS levels detected for these complexes and the 5B3A ligand (Table 7 and Figure 9, Appendix A). Our findings serve as another layer of evidence supporting the notion that elevated levels of ROS can promote mitochondrial impairment [106,107,108,109,110]. Furthermore, the RFU values reported for **P-5B3A**, **5-5B3A** and **56-5B3A**, and their equivalent platinum(II) precursors (**PHEN*SS***, **5ME*SS*** and **56ME*SS***) were almost equivalent at 24, 48 and 72 h (Table 8 and Figure 10 and Appendix A). This suggests that the coordinated 5B3A ligand did not play a significant role in altering MtMP. The changes in MtMP detected were predominantly caused by the core platinum(II) precursors (**PHEN*SS***, **5ME*SS*** and **56ME*SS***) of the complexes. The results indicate that the platinum(II) precursors of the studied platinum(IV) prodrugs produced more pronounced impairment in the mitochondria. This phenomenon may be linked to the reduction potential of the prodrugs in the mitochondria and to their mechanism of action with a comparable delayed response to that of their platinum(II) precursors. A more pronounced effect of the prodrugs may be observed at a timeframe of 96 h, possibly reaching the same potential as their platinum(II) precursors in causing mitochondrial impairment. 

#### 2.3.4. Histone Deacetylase (HDAC) Inhibition Studies

Histone acetyltransferases (HATs) and histone deacetylases (HDACs) are critical transcriptional enzymes that play a role in regulating gene expression [111]. Histones are chromatin-regulating proteins which serve as scaffolding that maintain the structural integrity of chromosomal DNA [112]. HATs are responsible for adding acetyl (−COCH_3_) to the highly charged lysine residues of histones [113]. This process eliminates the charge of the lysine residues, which decreases the affinity between histones and the DNA. As a result, the chromatin conforms into a relaxed state and this promotes gene activation by enhancing transcription [114]. Conversely, the primary function of HDACs is to remove acetyl from lysine residues of hyperacetylated histones and this forces the chromatin to condense, thus preventing further transcription [115].

Interestingly, HDACs are overexpressed in multiple cancer types and this overexpression aids in cell proliferation and survival [116]. For this reason, HDACs are considered valuable therapeutic targets in cancer therapy. Over the years, the development of HDAC inhibitors (HDACis) for cancer treatment has been a fruitful area of research [117,118]. To date, vorinostat, panobinostat, belinostat, romidepsin and chidamide are the only HDACis currently marketed for cancer treatment [119]. Several HDACis are also in clinical trials [120]. Other clinically investigated HDACis also include smaller molecules like valproic acid (VPA) [121,122] and 4-phenylburytic acid (PhB) [123], as well as a range of indole-based derivatives [124,125]. Of further note, HDACis are reported to reverse the tumour-related phenomenon, Warburg effect [126,127]. Unfortunately, treatment with HDACis alone have limited success in cancer therapy due to their poor pharmacokinetics (i.e., limited absorption, distribution, metabolism and excretion) [128], susceptibility to resistance [129,130], and low potency towards solid tumours [131]. Combination therapy is proposed to achieve the full potential of HDACis [132,133]. Another approach that can be employed is by coordinating HDACis to a platinum(IV) scaffold. For example, PhB has been coordinated to platinum(IV) complex with a cisplatin core which produced derivatives with enhanced antiproliferative effects and are capable of overcoming resistance [48,134]. This demonstrated a synergism between PhB and cisplatin [48]. At a molecular level, HDACis (i.e., PhB) can expose the DNA to platination, meaning that the affinity of DNA-binding agents (i.e., cisplatin) to DNA is increased [135].

We determined the inhibition of HDAC activity by **P-5B3A**, **5-5B3A** and **56-5B3A**, 5B3A, the equivalent platinum(II) precursors (**PHEN*SS***, **5ME*SS*** and **56ME*SS***) and platinum(IV) scaffolds (**PHEN*SS*(IV)(OH)_2_**, **5ME*SS*(IV)(OH)_2_** and **56ME*SS*(IV)(OH)_2_**), together with cisplatin in HT29 colon cells at 72 h (Table 9 and Figure 11 and Appendix A). All values are expressed as RFU. Relative to the HT29 extract (control), a lower RFU value corresponds to a decline in HDAC activity.

Peak HDAC inhibition was demonstrated by the potent HDAC inhibitor, TSA with a reported RFU value of 749 (Table 9 and Figure 11 and Appendix A). The studied platinum(IV) complexes, **P-5B3A**, **5-5B3A** and **56-5B3A**, the 5B3A ligand, the equivalent platinum(II) precursors (**PHEN*SS***, **5ME*SS*** and **56ME*SS***), the platinum(IV) scaffolds (**PHEN*SS*(IV)(OH)_2_**, **5ME*SS*(IV)(OH)_2_** and **56ME*SS*(IV)(OH)_2_**), and cisplatin exhibited considerable HDAC inhibition in HT29 colon cells at 72 h, relative to the HT29 extract (control) (Table 9 and Figure 11 and Appendix A). Interestingly, **PHEN*SS***, **PHEN*SS*(IV)(OH)_2_**, **P-5B3A**, the 5B3A ligand and cisplatin caused HDAC inhibition to a similar degree, with minor differences in the reported RFU values (Table 9). In comparison, **5ME*SS***, **56ME*SS***, **5ME*SS*(IV)(OH)_2_**, **56ME*SS*(IV)(OH)_2_**, **5-5B3A** and **56-5B3A** displayed greater HDAC inhibition when compared to their counterparts and cisplatin (Table 9 and Figure 11 and Appendix A). Noticeably, the complexes bearing the methylated heterocyclic ligands (**5ME*SS***, **56ME*SS***, **5ME*SS*(IV)(OH)_2_**, **56ME*SS*(IV)(OH)_2_**, **5-5B3A** and **56-5B3A**) are more potent HDAC inhibitors than the unmethylated complexes (**PHEN*SS***, **PHEN*SS*(IV)(OH)_2_** and **P-5B3A**) (Table 9 and Figure 11 and Appendix A). This pattern of HDAC inhibition further corroborates that methylation of the heterocyclic ligand is critical to overall biological activity. Moreover, this pattern can also be associated with the increase in ROS levels (Table 7 and Figure 9) and decrease in mitochondrial activity (Table 8 and Figure 10) observed with these complexes.

Surprisingly, **5ME*SS*** and **56ME*SS*** displayed the greatest HDAC inhibition amongst all the platinum complexes in this study with reported RFU values of 2124 and 1847 (Table 9), respectively. It is not clear why the other platinum(II) derivative, **PHEN*SS*** was not able to cause the same degree of HDAC inhibition. However, it may be implied that the methylation at the heterocyclic ligands of **5ME*SS*** and **56ME*SS*** may have aided the complexes to have better inhibitory response towards HDAC. We also believe that this phenomenon may also be related to the mode of action of the complexes. Although the mode of action of **5ME*SS*** has not been investigated before, we can only speculate that it parallels the proposed mode of action of **56ME*SS*** due to their structural similarity. **56ME*SS*** is reported to interfere with cytoskeletal networks [36,37]. Briefly, there are 18 HDACs present in humans [115], and one of them is a cytoplasmic enzyme called HDAC-6 that is responsible for regulating cytoskeletal assembly [136,137,138]. HDAC-6 is overexpressed in HT29 colon cells [139,140]. Based on this information, it may be concluded that the strong inhibitory response of **56ME*SS*** towards HDAC is a consequence of its ability to inhibit the cytoskeleton. Of further note, while it was anticipated that the platinum(IV) complexes, **5-5B3A** and **56-5B3A** would display better HDAC inhibition than their platinum(II) counterparts this was not reflected in the results. The weaker HDAC inhibitory response of **5-5B3A** and **56-5B3A** may be due to non-additive or antagonistic effect with the coordinated ligand, 5B3A.

## 3. Materials and Methods

### 3.1. Materials

All reagents used were of spectroscopic grade. The deionised water (d.i.H_2_O) used in the experimentations was acquired from a MilliQ^TM^ system (Millipore Australia Pty Ltd., Sydney, NSW, Australia). Phen, 5-Mephen, 5,6-Me_2_phen, 5B3A, acetonitrile (CH_3_CN), dimethyl sulfoxide (DMSO), N-hydroxysuccinimide, AsA, *N*,*N′*-dicyclohexylcarbodiimide (DCC) and trifluoroacetic acid (TFA) were purchased from Sigma-Aldrich, Sydney, NSW, Australia. Deuterated DMSO (DMSO-d_6_) was purchased from Cambridge Isotope Laboratories, Andover, MA, USA. Methanol (MeOH) was obtained from Honeywell Research Chemicals, NJ, USA. Acetone (C_3_H_6_O) and diethyl ether (Et_2_O) were purchased from ChemSupply Australia, Gillman, SA, Australia. Additional reagents used in the experiments were purchased from commercial sources. The cell lines HT29 colon, U87 glioblastoma, MCF-7 breast, H460 lung, Du145 prostate, BE2-C neuroblastoma, MIA pancreas, the cisplatin-resistant ADDP ovarian variant and the non-tumour-derived MCF10A breast were sourced from the American Type Culture Collection (ATCC). The A431 skin and A2780 ovarian cell lines were purchased from the European Collection of Authenticated Cell Cultures (ECACC). The SJ-G2 glioblastoma was provided by Dr. Mary Danks of St. Jude Children’s Research Hospital, Memphis, TN, USA. All cell lines were authenticated by CellBank Australia, (part of Children’s Medical Research Institute, Westmead, Sydney, NSW, Australia).

### 3.2. Chemistry

#### 3.2.1. Synthesis Route of NHS-5B3A

NHS-5B3A was prepared according to established protocols [60]. 5B3A was reacted with 1 mol eq. of NHS and DCC in C_3_H_6_O. The reaction solution was left stirring at room temperature for 24 h. The precipitated by-product, dicyclohexylurea (DCU), was removed through syringe filtration. The filtrate was collected and reduced to dryness through rotary evaporation. The NHS ester was used without further purification.

**NHS-5B3A**—Yield: 320 mg; 84%. ^1^H-NMR (400 MHz, DMSO-d_6,_ δ): 10.9 (s, H_pyrrole_, 1H), 7.48 (d, a and b, 2H, *J* = 7.6 Hz), 7.40 (t, c and d, 2H, *J* = 7.6 Hz), 7.33 (m, e and f, 2H, *J* = 7.8 Hz), 7.28 (d, i, 1H, *J* = 8.8 Hz), 7.18 (s, g, 1H), 6.83 (dd, h, 1H, *J*_1_
*=* 1.9 Hz, *J*_2_
*=* 8.7 Hz), 5.10 (s, β, 2H), 4.13 (s, α, 2H), and 2.83 (s, j and k, 4H). HPLC λ_max_ nm, T_R_: 254 nm, 11.8 min.

#### 3.2.2. Syntheses of Platinum(II) Precursors and Platinum(IV) Scaffolds

All platinum(II) precursors of type **[Pt^II^(H_L_)(A_L_)]^2+^** and platinum(IV) scaffolds of type **[Pt^IV^(H_L_)(A_L_)(OH)_2_]^2+^** were prepared as reported [59].

#### 3.2.3. Syntheses of Platinum(IV) Derivatives Incorporating 5B3A (**P-5B3A**, **5-5B3A** and **56-5B3A**)

**[Pt^IV^(H_L_)(A_L_)(X)(OH)]^2+^** were synthesised according to established protocols [61]. The appropriate precursor, **[Pt^IV^(H_L_)(A_L_)(OH)_2_]^2+^**, was reacted with 2 mol eq. of NHS ester of 5B3A in DMSO (1–2 mL) for 72 h at room temperature, in the dark. The reaction solution was washed with excess Et_2_O and vigorously mixed using a plastic pipette, followed by centrifugation to afford a colourless supernatant and an oily brown layer. The colourless supernatant was discarded while the oily brown layer was collected and dissolved in methanol (1–2 mL), followed by the addition of excess Et_2_O to induce precipitation. Centrifugation was undertaken to collect the final precipitate. Excess C_3_H_6_O was mixed with the precipitate and sonicated, affording a pure and solidified yellow or green precipitate that was collected through centrifugation.

**P-5B3A**—Yield: 64 mg; 80%. ^1^H-NMR (400 MHz, DMSO-d_6,_ δ): 10.3 (s, H_pyrrole_, 1H), 9.40 (d, H2, 1H, *J* = 5.4 Hz), 9.38 (d, H9, 1H, *J* = 5.4 Hz), 9.06 (dd, H4 and H7, 2H, *J*_1_
*=* 4.8 Hz, *J*_2_ = 8.1 Hz), 8.26 (s, H5 and H6, 2H), 8.32 (m, H3 and H8, 2H), 7.51 (d, a and b, 2H, *J =* 7.5 Hz), 7.45 (t, c and d, 2H, *J =* 7.4 Hz), 7.38 (t, e, 1H, *J =* 7.1 Hz), 7.02 (d, i, 1H, *J =* 8.7 Hz), 6.69 (dd, h, 1H, *J*_1_
*=* 2.0 Hz, *J*_2_
*=* 8.6 Hz), 6.46 (s, f and g, 2H), 4.91 (s, β, 2H), 3.23 (d, α, 2H, *J =* 6.4 Hz), 2.98 (s, H1′ and H2′, 2H), 2.23 (m, H3′ and H6′ eq., 2H), 1.64 (m, H4′ and H5′ eq.; H3′ and H6′ ax., 4H), and 1.23 (m, H4′ and H5′ ax., 2H). ^1^H-^195^Pt-HMQC (400 MHz, DMSO-d_6,_ δ): 9.40/498 ppm; 9.38/498 ppm; 8.32/498 ppm; 3.24/498 ppm. HPLC λ_max_ nm, T_R_: 254 nm, 8.08 min. UV λ_max_ nm (ε/M.cm^−1^ ± SD × 10^4^, d.i.H_2_O): 304 (0.73 ± 3.15), 279 (2.16 ± 1.63), 204 (8.08 ± 1.70). CD λ_max_ nm (Δε/M.cm^−1^ × 10^1^, d.i.H_2_O): 202 (−463), 203 (−485), 205 (−551), 228 (−58.9), 239 (−99.6), 263 (+31), 269 (+5.15), 274 (+21.3), 285 (−16.5), 313 (−32.2). Elemental microanalysis: *calculated* for C_35_H_37_N_7_O_10_Pt: C: 44.40%; H: 4.36%; N: 10.36%, found: C: 44.12%; H: 4.38%, N: 10.15%. Solubility: 7 mg/mL.

**5-5B3A**—Yield: 76 mg; 77%. ^1^H-NMR (400 MHz, DMSO-d_6,_ δ): 10.3 (d, H_pyrrole_, 1H, *J =* 1.8 Hz), 9.33 (m, H2 and H9, 2H), 9.05 (dd, H4, 1H, *J =* 7.8 Hz), 8.91 (dd, H7, 1H, *J*_1_
*=* 4.4 Hz, *J*_2_
*=* 8.2 Hz), 8.30 (m, H3 and H8, 2H), 7.52 (d, a and b, 2H, *J =* 7.2 Hz), 7.46 (t, c and d, 2H, *J =* 7.3 Hz), 7.39 (t, e, 1H, *J =* 7.2 Hz), 7 (d, i, 1H, *J =* 8.7 Hz), 6.68 (d, h, 1H, *J =* 8.7 Hz), 6.42 (d, f, 1H, *J =* 2.2 Hz), 6.38 (s, g, 1H), 4.86 (s, β, 2H), 3.20 (m, α, 2H), 2.99 (s, H1′ and H2′, 2H), 2.88 (s, CH_3_, 3H), 2.24 (m, H3′ and H6′ eq., 2H), 1.66 (m, H4′ and H5′ eq.; H3′ and H6′ ax., 4H), and 1.25 (m, H4′ and H5′ ax., 2H). ^1^H-^195^Pt-HMQC (400 MHz, DMSO-d_6,_ δ): 9.33/499 ppm; 9.05/499 ppm; 8.91/499 ppm; 3.20/499 ppm. HPLC λ_max_ nm, T_R_: 254 nm, 8.38 min. UV λ_max_ nm (ε/M.cm^−1^ ± SD × 10^4^, d.i.H_2_O): 310 (1.01 ± 0.73), 286 (3.63 ± 0.33), 204 (13.5 ± 3.61). CD λ_max_ nm (Δε/M.cm^−1^ × 10^1^, d.i.H_2_O): 202 (−433), 203 (−460), 204 (−490), 230 (−35.1), 240 (−65.1), 257 (+16.9), 263 (+28.6), 270 (+30.7), 286 (−53.5). Elemental microanalysis: *calculated* for C_36_H_39_N_7_O_10_Pt: C: 44.17%; H: 4.63%; N: 10.02%, found: C: 44.22%; H: 4.44%, N: 9.92%. Solubility: 6 mg/mL.

**56-5B3A**—Yield: 58 mg; 88%. ^1^H-NMR (400 MHz, DMSO-d_6,_ δ): 10.3 (d, H_pyrrole_, 1H, *J =* 1.9 Hz), 9.31 (d, H2, 1H, *J* = 5.4 Hz), 9.25 (d, H9, 1H, *J* = 5.4 Hz), 9.08 (dd, H4 and H7, 2H, *J* = 7.9 Hz), 8.27 (m, H3 and H8, 2H), 7.52 (d, a and b, 2H, *J =* 7.1 Hz), 7.47 (t, c and d, 2H, *J =* 7.3 Hz), 7.42 (t, e, 1H, *J =* 7.2 Hz), 6.95 (d, i, 1H, *J =* 8.7 Hz), 6.64 (dd, h, 1H, *J*_1_
*=* 2.4 Hz, *J*_2_
*=* 8.7 Hz), 6.36 (d, f, 1H, *J =* 2.2 Hz), 6.30 (d, g, 1H, *J =* 2.2 Hz), 4.80 (s, β, 2H), 3.18 (d, α, 2H, *J =* 4.8 Hz), 2.99 (m, H1′ and H2′, 2H), 2.79 (m, 2 × CH_3_, 6H), 2.25 (d, H3′ and H6′ eq., 2H), 1.67 (m, H4′ and H5′ eq.; H3′ and H6′ ax., 4H), and 1.25 (m, H4′ and H5′ ax., 2H). ^1^H-^195^Pt-HMQC (400 MHz, DMSO-d_6,_ δ): 9.31/485 ppm; 9.25/485 ppm; 3.18/485 ppm. HPLC λ_max_ nm, T_R_: 254 nm, 8.54 min. UV λ_max_ nm (ε/M.cm^−1^ ± SD × 10^4^, d.i.H_2_O): 315 (0.59 ± 0.94), 291 (2.02 ± 1.78), 250 (1.31 ± 2.53), 206 (8.22 ± 1.75). CD λ_max_ nm (Δε/M.cm^−1^ × 10^1^, d.i.H_2_O): 202 (−376), 204 (−384), 208 (−425), 230 (−6.55), 235 (−25.4), 239 (+7.34), 248 (+73.4), 257 (+101), 275 (+80.7), 288 (+0.91). Elemental microanalysis: *calculated* for C_37_H_41_N_7_O_10_Pt: C: 45.58%; H: 4.65%; N: 10.06%, found: C: 45.77%; H: 4.28%, N: 10.33%. Solubility: 4 mg/mL.

### 3.3. Laboratory Instrumentation

Equipment utilised to verify the purity and confirm the chemical structures of the studied platinum(IV) complexes including relevant protocols are outlined below.

#### 3.3.1. Flash Chromatography

A Biotage Isolera^TM^ One flash chromatography system (Shimadzu, Sydney, NSW, Australia) equipped with a Biotage^®^ Sfär C18 D (Duo 100 Å 30 μm 30 g) (Shimadzu, Sydney, NSW, Australia) was employed to purify the platinum(IV) complexes. The mobile phase consisted of solvents, A (d.i.H_2_O) and B (MeOH). The samples were dissolved in d.i.H_2_O/MeOH (50:50) and eluted through the column with a 0–40% linear gradient for 40 min with a flow rate of 4 mL.min^−1^, collected within the set wavelengths of 200–400 nm.

#### 3.3.2. High-Performance Liquid Chromatography (HPLC)

An Agilent (Melbourne, VIC, Australia) Technologies 1260 Infinity instrument equipped with a Phenomenex Onyx^TM^ Monolithic C_18_ reverse-phase column (100 × 4.6 mm, 5 µm pore size) (Sydney, NSW, Australia) was utilised for the complexes. The mobile phase consisted of solvents, A (0.06% TFA in d.i.H_2_O) and B (0.06% TFA in CH_3_CN/d.i.H_2_O (90:10)). An injection volume of 5 µL was utilised and eluted with a 0–100% linear gradient over 15 min with a flow rate of 1 mL.min^−1^, at the set wavelengths of 214 and 254 nm. An Agilent ZORBAX RX-C_18_ column (100 × 4.6 mm, 3.5 µm pore size) (Sydney, NSW, Australia) was utilised for the NHS ester of 5B3A using the same method described above.

#### 3.3.3. Nuclear Magnetic Resonance (NMR) Spectroscopy

^1^H-NMR, 2D-COSY, ^1^H-^195^Pt-HMQC and 1D-^195^Pt-NMR were carried out on a 400 MHz Bruker (Melbourne, VIC, Australia) Avance spectrometer at 298 K. All complexes were prepared to a concentration of 10 mM in 600 µL using D_2_O. DMSO-d_6_ was utilised for 5B3A, NHS ester of 5B3A, and the platinum(IV) complexes, **P-5B3A, 5-5B3A** and **56-5B3A**. ^1^H-NMR was set to 10 ppm and 16 scans with a spectral width of 8250 Hz and 65,536 data points. 2D-COSY was acquired using a spectral width of 3443 Hz for both ^1^H nucleus, F1 and F2 dimensions, with 256 and 2048 data points, respectively. ^1^H-^195^Pt-HMQC was carried out using a spectral width of 214,436 Hz and 256 data points for ^195^Pt nucleus, F1 dimension, also a spectral width of 4808 Hz with 2048 data points for ^1^H nucleus, F2 dimension. 1D-^195^Pt was measured using a spectral width of 85,470 Hz and 674 data points. All resonance recorded, were presented as chemical shifts in parts per million (δ ppm) with *J*-coupling constants reported in Hz. For spin multiplicity: s (singlet); d (doublet); dd (doublet of doublets); t (triplet); q (quartet) and m (multiplet). All spectroscopic data gathered were generated and plotted using TopSpin 4.1.3 analysis software.

#### 3.3.4. Ultraviolet–Visible (UV) Spectroscopy

An Agilent (Melbourne, VIC, Australia) Technologies Cary 3500 UV–Vis Multicell Peltier spectrophotometer was utilised to perform the UV spectroscopy experiments. UV spectroscopic experiments were completed at room temperature in the range of 200–400 nm with a 1 cm quartz cuvette. All complexes were prepared in d.i.H_2_O, while 5B3A was prepared in CH_3_CN. A stock solution of each complex or compound (1 mM) was prepared, and absorption spectra were recorded at a series of different concentrations by titrating 9 × 3 µL aliquots into a cuvette containing d.i.H_2_O (3000 µL). Experiments were repeated in triplicate. All spectra were baseline corrected by the instrument—a baseline containing d.i.H_2_O was acquired first, and automatically subtracted from each experiment. Average extinction coefficients (ε) were determined with standard deviation and errors based on the generated plot curves.

#### 3.3.5. Circular Dichroism (CD) Spectroscopy

A Jasco (Easton, PA, USA) J-810 CD spectropolarimeter was used to measure the CD spectra of the complexes. The samples were prepared in d.i.H_2_O, with similar concentrations (~0.05 mM) using a 1 mm optical glass cuvette or 1 cm quartz cuvette. CD experiments were undertaken at room temperature in the wavelength range of 200–400 nm (30 accumulations) with a bandwidth of 1 nm, data pitch of 0.5 nm, a response time of 1 s and a 100 nm.min^−1^ scan speed. The flowrate of nitrogen gas was 6 L.min^−1^. The HT (photomultiplier) level remained below 500 V for all experiments. A CD simulation tool (CDToolX—Windows 10 Version) was used to process the spectra.

#### 3.3.6. Elemental Microanalysis

Elemental analysis was performed on Elementar Vario MICRO (Elementar Analysensysteme GmbH, Frankfurt, Germany). The instrument hardware was configured for the analysis of 4 elements (C, H, N and S). An amount of 1.2 ± 0.2 mg of sample material was loaded into a tin foil crucible and combusted at 1150 °C with oxygen dosing time of 80 s and total O_2_ flow rate of 30 mL/min. Ultra-high purity grade helium (BOC, 99.999%) and oxygen (BOC, 99.995%) were employed as working fluids in all cases. Pure sulfanilamide was employed as a standard with quality control samples added into the workflow every 10–20 runs. The follow-up data analysis was performed using a custom peak picking and integration algorithm written in Python 3.11.

### 3.4. Physicochemical Measurements

#### 3.4.1. Solubility

The solubility of **P-5B3A**, **5-5B3A** and **56-5B3A** was tested in d.i.H_2_O at room temperature. Small aliquots of d.i.H_2_O were titrated into an Eppendorf tube containing 1 mg of each complex until full dissolution. For every titration, each sample solution was vortexed and sonicated.

#### 3.4.2. Stability 

The stability of **P-5B3A**, **5-5B3A** and **56-5B3A** were measured at 254 nm. Each complex was incubated in aqueous phosphate-buffered saline (PBS) solution (pH ~7.4) (total sample concentration, ~10 mM) at 37 °C for one week. Analytical HPLC was utilised to monitor the experiments.

#### 3.4.3. Lipophilicity

Lipophilicity measurements were completed via analytical HPLC, as previously reported [61].

#### 3.4.4. Reduction

The reduction behaviour of **P-5B3A**, **5-5B3A** and **56-5B3A** was investigated using ^1^H-NMR and 1D-^195^Pt-NMR spectroscopy, using previously established methods [61].

### 3.5. Biological Measurements

#### 3.5.1. Cell Viability Assays

Cell viability assays were completed at the Calvary Mater Newcastle Hospital, NSW, Australia, as previously described [61,72]. The MTT (3-[4,5-dimethylthiazol-2-yl]-2,5-diphenyltetrazolium bromide) assay was used to measure the drug concentration at which cell growth was inhibited by 50% or GI_50_. **P-5B3A**, **5-5B3A**, **56-5B3A** and 5B3A were tested against the following cell lines: HT29 colon, U87 glioblastoma, MCF-7 breast, A2780 ovarian, H460 lung, A431 skin, Du145 prostate, BE2-C neuroblastoma, SJ-G2 glioblastoma, MIA pancreas, the cisplatin-resistant ADDP ovarian variant, and the non-tumour-derived MCF10A breast line. The SCI for **P-5B3A**, **5-5B3A** and **56-5B3A** was also calculated by dividing the GI_50_ values of the complexes in the normal breast cell line MCF10A by their GI_50_ in all cancerous cell lines. Generally, the higher the SCI the more selective a drug is towards cancer cells [74,75].

#### 3.5.2. Drug-Induced Reactive Oxygen Species (ROS) Production

To evaluate the presence of ROS in treated cells, a DCFDA/H_2_DCFDA-cellular ROS Assay Kit (Abcam, Cambridge, MA, USA) was used, as previously described [61,141,142]. A total of 25,000 cells/mL of HT29 cells in DMEM were seeded in 96-well plates. Cells were washed with 1X kit buffer, and then stained with 25 μM 2′,7′-dichlorofluorescein diacetate (DCFH-DA) and incubated for 45 min. DCFH-DA was then removed, and cells were then re-washed with 1X kit buffer, after which phenol red free media was added. Cells were then treated with a GI_50_ drug concentration for each complex. The plates were directly scanned to measure fluorescence (relative fluorescence units (RFU)) at the time points indicated the Glo-Max^®^-Multimode microplate reader (Promega Corporation, Alexandra, VIC, Australia) at an excitation/emission of 485/535 nm, which is directly correlated to the DCFH-DA that is deacetylated by cellular esterases and then oxidised by ROS to the fluorescent compound DCF. To generate the positive control (20 μM *tert*-butyl hydroperoxide (TBHP)), cells were washed with 1X kit buffer, and stained with DCFDA (25 μM) for 45 min; this was removed and TBHP was added in phenol red free media, and the resulting solution was scanned as described above.

#### 3.5.3. Mitochondrial Membrane Potential (MtMP) Changes

To investigate the MtMP changes in HT29 treated cells, a TMRE-MtMP Assay Kit (Abcam, Cambridge, MA, USA) was used, as previously described [36,61]. In 96-well plates, 25,000 cells/mL of HT29 colon cells in DMEM were seeded. Cells were treated with a GI_50_ drug concentration for each complex. At 24, 48 or 72 h, cells were washed with PBS and stained with tetramethylrhodamine, ethyl ester (TMRE) (1 μM) and incubated for 30 min. TMRE was then removed, and cells were then re-washed with PBS (0.2% BSA), after which phenol red free media was added. The plates were directly scanned to measure fluorescence (expressed as RFU), using the Glo-Max^®^-Multimode microplate reader (Promega Corporation, Alexandra, VIC, Australia) at an excitation/emission of 549/575 nm. For the positive control, 20 μM carbonyl cyanide 4-(trifluoromethoxy) phenylhydrazone (FCCP)) was added to the cells and incubated for 10 min. The cells were washed with PBS (0.2% BSA) and stained with TMRE (1 μM) for 30 min; this was removed, and phenol red free media was added, and the final solution was scanned as mentioned above.

#### 3.5.4. Histone Deacetylase (HDAC) Inhibition Measurements

To study the HDAC activity in HT29 treated cells, a Fluorometric HDAC Activity Assay Kit ab156064 (Abcam, Cambridge, MA, USA) was used and all sample preparations were made based on the manufacturer’s protocol. A total of 10^6^ cells/mL of HT29 cells were seeded in DMEM with serum deprivation for 18 h to synchronise the cells. Then, cells were treated with GI_50_ drug concentration of each complex or compound for 72 h. Cells were then washed with cold PBS and resuspended in lysis buffer (10 mM Tris HCl (pH 7.5), 10 mM NaCl, 15 mM MgCl_2_, 250 mM sucrose, 0.5% NP-40 and 0.1 mM EGTA) and lysed on ice for 20 min. Cells were centrifuged through 4 mL of sucrose cushion (30% sucrose, 10mM Tris HCl (pH 7.5), 10 mM NaCl and 3mM MgCl_2_) at 1300× *g* for 10 min at 4 °C. Supernatant was discarded and the pellet was washed and resuspended in extraction buffer (50 mM HEPES-KOH (pH 7.5), 420 mM NaCl, 0.5 mM EDTA 2Na, 0.1 mM EGTA and 10% glycerol) and then sonicated for 30 s. Nuclei were lysed for 30 min on ice and centrifuged at 20,000× *g* for 30 min. The supernatant was collected, and the samples were prepared for the fluorometric assay according to the manufacturer’s protocol (Histone Deacetylase (HDAC) Activity Assay Fluorometric Kit ab156064, Abcam). The changes in the relative fluorescence intensity (expressed as RFU) were recorded each minute over 60 min using the Glo-Max^®^-Multimode microplate reader (Promega Corporation, Alexandra, VIC, Australia) at an excitation/emission of 355/460 nm.

## 4. Conclusions

Three platinum(IV) complexes incorporating the indole-based ligand, 5B3A (**P-5B3A**, **5-5B3A** and **56-5B3A**) were successfully synthesised in good yields. HPLC, NMR, UV and CD, as well as elemental microanalysis experiments confirmed the purity and chemical structures of the resultant complexes. The reduction behaviour of the complexes in PBS and AsA were monitored via NMR at 37 °C, with results indicating the ability to reduce to the equivalent platinum(II) precursors. From this, it is proposed that they can be defined as prodrugs. Additionally, the complexes were adequately stable in PBS for up to one week at 37 °C, suggesting their resistance towards significant hydrolysis in aqueous solutions. The rank of increasing lipophilicity of the complexes, as determined by analytical HPLC, was **P-5B3A** < **5-5B3A** < **56-5B3A**.

The MTT assay was used to measure the anticancer activity of **P-5B3A**, **5-5B3A** and **56-5B3A**. The findings indicated that the complexes were considerably more potent than cisplatin, oxaliplatin and carboplatin in most cell lines tested. This means that the coordination of the 5B3A ligand to our platinum(IV) scaffolds enhanced the biological activity of the resultant platinum(IV) complexes to an extent. The rank of increasing cytotoxicity was determined as **P-5B3A** < **5-5B3A** < **56-5B3A**. This also relates with the rank of increasing lipophilicity. Notably, the most lipophilic complex, **56-5B3A** displayed the greatest growth inhibitory response in the Du145 prostate and HT29 colon cell lines with reported GI_50_ values of 1.2 ± 0.6 nM and 4.4 ± 1.6 nM, respectively. Unfortunately, **P-5B3A**, **5-5B3A** and **56-5B3A** were found to be significantly toxic towards the MCF10A normal breast cell line, denoting their limited selectivity. While this is the case, it should be noted that this outcome may not be translated in vivo. Despite this, the complexes were still significantly more potent than cisplatin in the cisplatin-resistant ADDP ovarian variant cell lines, thus confirming their ability to overcome resistance mechanisms by cancer cells.

A sustained increase in ROS production was also observed in HT29 colon cells, upon treatment with **P-5B3A**, **5-5B3A** and **56-5B3A** for up to 72 h. The order by which the complexes displayed cumulative ROS production at 24, 48 and 72 h was determined to be **P-5B3A** < **5-5B3A** < **56-5B3A**. This trend in ROS production also relates to the trend in cytotoxicity, further confirming that the presence of the 5B3A ligand in the axial position aided the complexes to produce ROS. Moreover, the observed ROS production was also in agreement with the decrease in mitochondrial activity induced by the complexes. **P-5B3A**, **5-5B3A** and **56-5B3A** caused a progressive decrease in MtMP in HT29 colon cells for up to 72 h, which translates to mitochondrial dysfunction. These results strongly suggest that ROS production and mitochondrial dysfunction via decreasing MtMP may be additional cytotoxic elements that contribute to the overall potency of the complexes.

The HDAC inhibition of **P-5B3A**, **5-5B3A** and **56-5B3A** was also investigated in HT29 colon cells. We anticipated that the 5B3A ligand would enhance the inhibitory response of the resultant complexes towards HDAC, because of its indole pharmacophore which has been reported to inhibit HDAC. Unexpectedly, the results indicated otherwise. **P-5B3A** proved to be the least active against HDAC. While the other derivatives, **5-5B3A** and **56-5B3A** displayed a better inhibitory response against HDAC, the degree of inhibition was significantly inferior compared to TSA. Clearly, the 5B3A ligand did not enhance the activity of **P-5B3A**, **5-5B3A** and **56-5B3A** against HDAC. For the first time, we also reported the HDAC inhibition of their equivalent platinum(II) precursors (**PHEN*SS***, **5ME*SS*** and **56ME*SS***) and platinum(IV) scaffolds (**PHEN*SS*(IV)(OH)_2_**, **5ME*SS*(IV)(OH)_2_** and **56ME*SS*(IV)(OH)_2_**) in the same cell line. Amongst the complexes, **5ME*SS*** and **56ME*SS*** proved somewhat active against HDAC. Particularly for **56ME*SS***, we speculate that its ability to inhibit HDAC significantly is closely associated with one of its mode of actions, that is its interference with the cytoskeleton.

## 5. Patents

This work is part of Australian PCT application (**PCT/AU2023/050027**), Platinum(IV) complexes, 20 February 2023, Western Sydney University, Sydney, Australia.

## Figures and Tables

**Figure 1 ijms-25-02181-f001:**
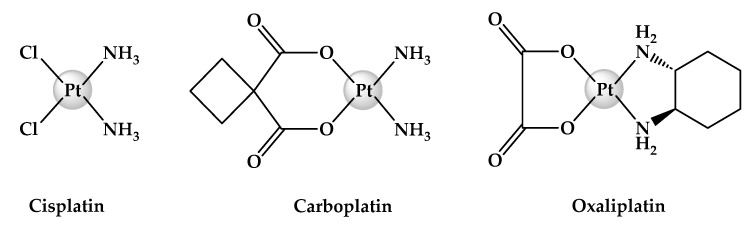
Molecular structures of standard platinum(II) drugs with worldwide approval for cancer treatment.

**Figure 2 ijms-25-02181-f002:**
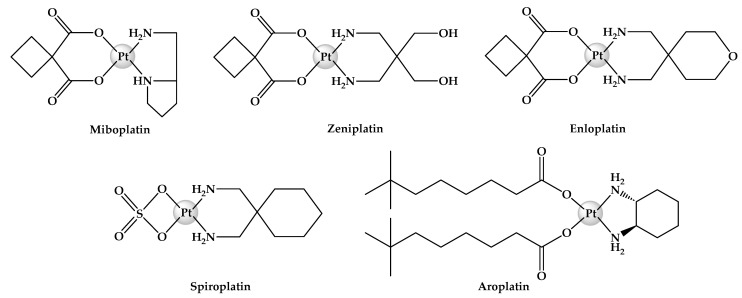
Molecular structures of third-generation platinum(II) complexes that were clinically investigated.

**Figure 3 ijms-25-02181-f003:**
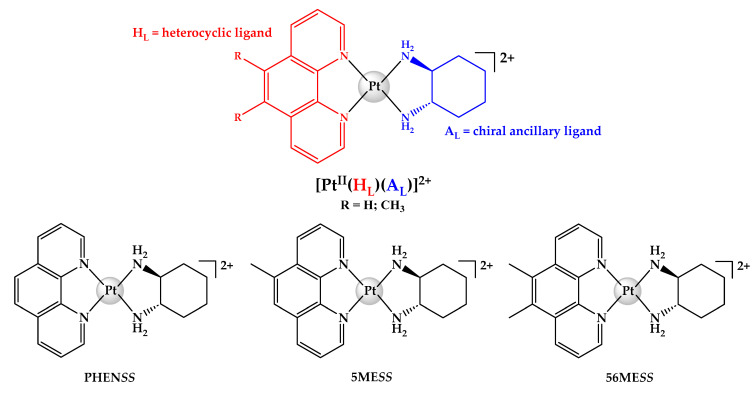
Molecular structures of unconventional platinum(II) complexes featuring a bulky heterocyclic ligand and a chiral ancillary ligand.

**Figure 4 ijms-25-02181-f004:**
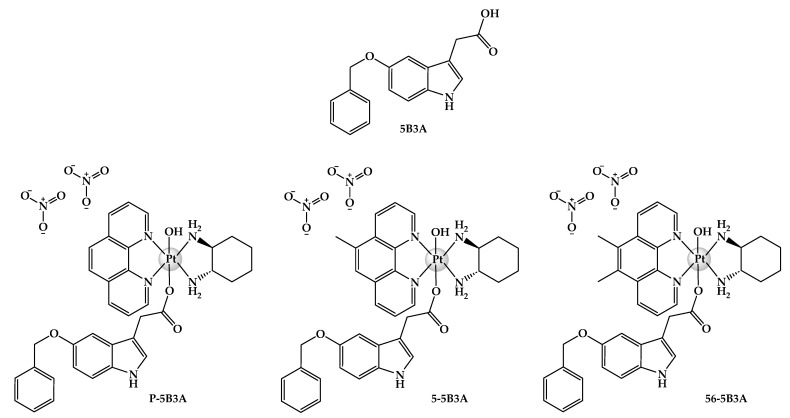
Molecular structures of the indole-based axial ligand, 5B3A, and the synthesised platinum(IV) complexes, **P-5B3A**, **5-5B3A** and **56-5B3A**.

**Figure 5 ijms-25-02181-f005:**
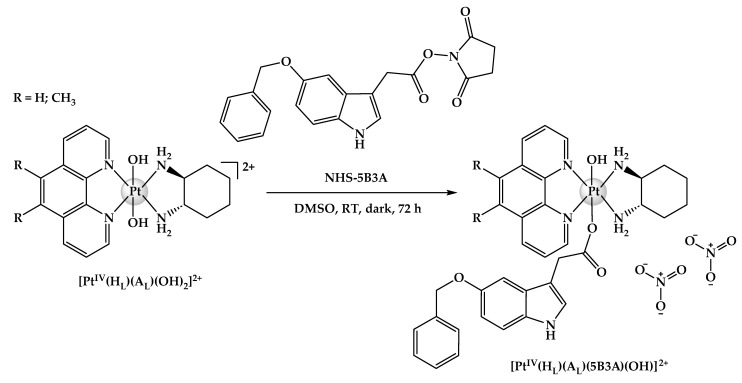
Synthetic scheme of **P-5B3A**, **5-5B3A** and **56-5B3A**.

**Figure 6 ijms-25-02181-f006:**
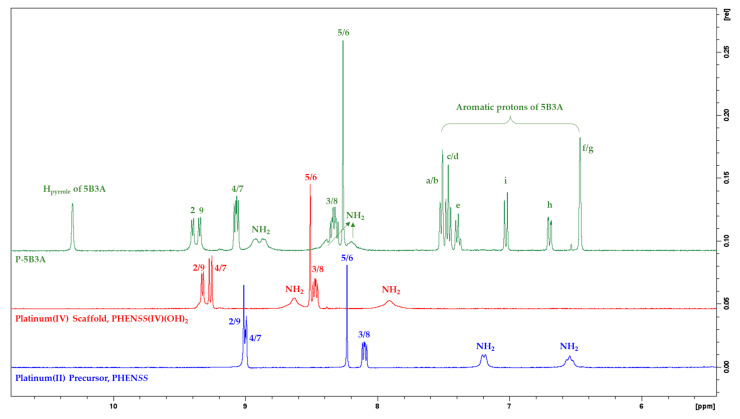
Expanded ^1^H-NMR spectra of **P-5B3A** and its equivalent platinum(II) precursor and platinum(IV) scaffold, **PHEN*SS*** and **PHEN*SS*(IV)(OH)_2_**, respectively, in DMSO-d_6_, showing the multiplicity variations and shifts in chemical resonances.

**Figure 7 ijms-25-02181-f007:**
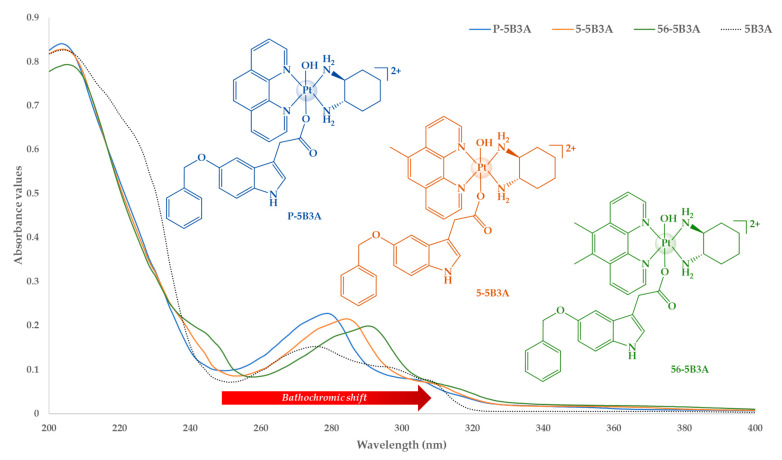
UV spectra of **P-5B3A**, **5-5B3A**, **56-5B3A** and the 5B3A ligand, obtained at 298 K, displaying the recorded UV absorption bands at various wavelengths. Inset: colour-coded structures of **P-5B3A**, **5-5B3A** and **56-5B3A**.

**Figure 8 ijms-25-02181-f008:**
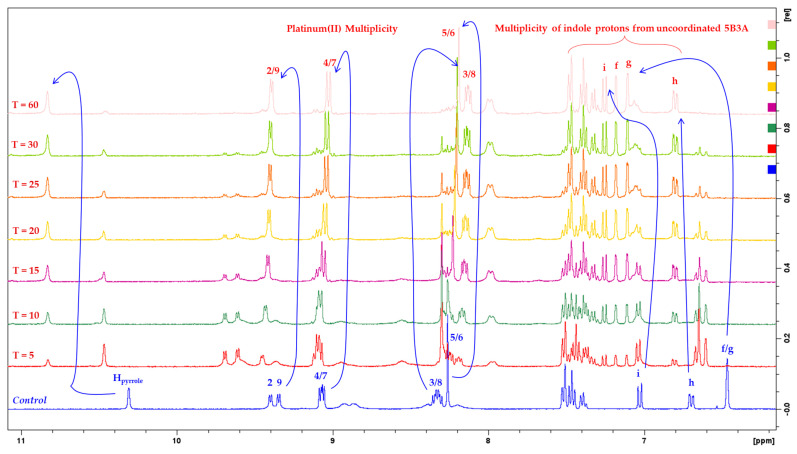
The expanded ^1^H-NMR spectra of **P-5B3A** treated with PBS and AsA in DMSO-d_6_ at 37 °C, highlighting the variations in chemical multiplicity in the aromatic region depicted by the arrows. **T** is time in min. The expanded ^1^H-NMR spectrum of the untreated complex (control) was also included for comparison.

**Figure 9 ijms-25-02181-f009:**
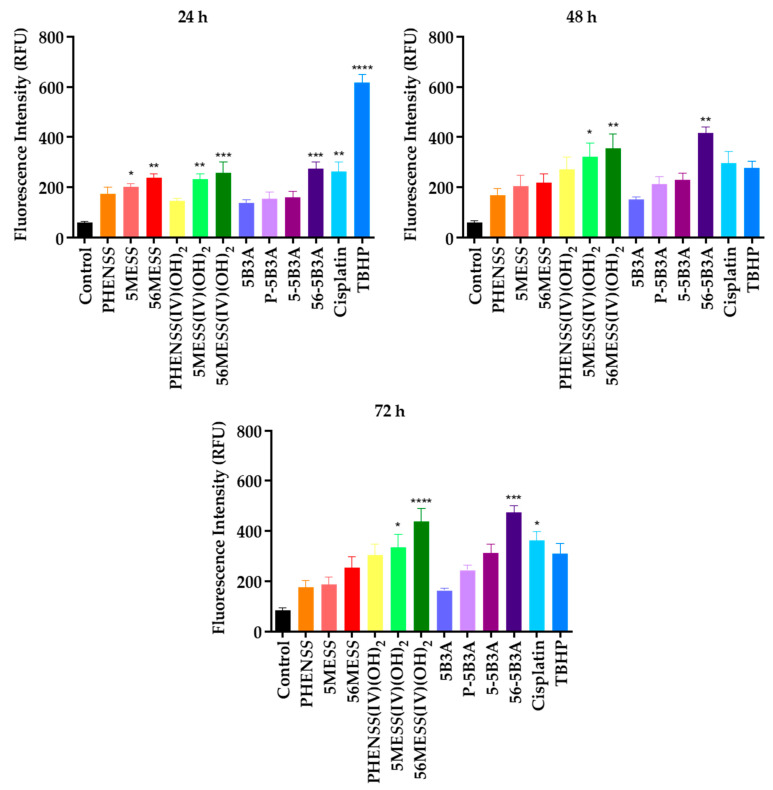
ROS generation in HT29 colon cells after treatment with **P-5B3A**, **5-5B3A** and **56-5B3A**, 5B3A, the equivalent platinum(II) precursors (**PHEN*SS***, **5ME*SS*** and **56ME*SS***) and platinum(IV) scaffolds (**PHEN*SS*(IV)(OH)_2_**, **5ME*SS*(IV)(OH)_2_** and **56ME*SS*(IV)(OH)_2_**), cisplatin and TBHP at 24, 48 and 72 h. Data are expressed in RFU values. The RFU values of the platinum(II) precursors (**PHEN*SS***, **5ME*SS*** and **56ME*SS***) and the platinum(IV) scaffolds (**PHEN*SS*(IV)(OH)_2_**, **5ME*SS*(IV)(OH)_2_** and **56ME*SS*(IV)(OH)_2_**) were included in the diagrams for comparison [59,61,65]. Data points signify the mean ± SEM. *n* = 3 from three independent tests where samples were achieved in triplicate. Significance was identified by Tukey’s multiple comparisons test for multiple comparisons. Group differences were considered statistically significant if * *p* < 0.05, ** *p* < 0.01, *** *p* < 0.001 and **** *p* < 0.0001 compared with control, as measured by one-way ANOVA.

**Figure 10 ijms-25-02181-f010:**
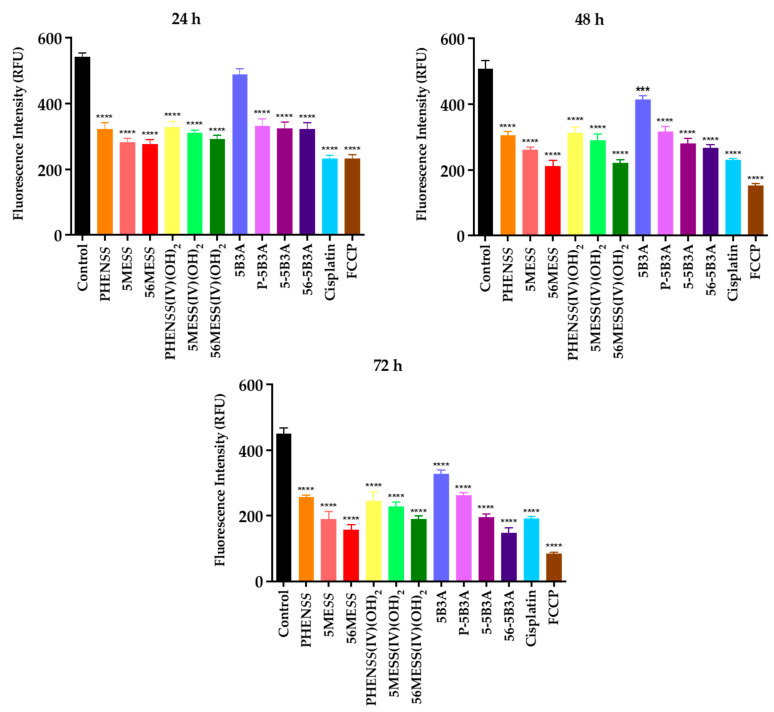
The detected MtMP changes in HT29 colon cells after treatment with **P-5B3A**, **5-5B3A** and **56-5B3A**, 5B3A, the equivalent platinum(II) precursors (**PHEN*SS***, **5ME*SS*** and **56ME*SS***) and platinum(IV) scaffolds (**PHEN*SS*(IV)(OH)_2_**, **5ME*SS*(IV)(OH)_2_** and **56ME*SS*(IV)(OH)_2_**), cisplatin and FCCP at 24, 48 and 72 h. Data are expressed in RFU values. Data points signify the mean ± SEM. *n* = 3 from three independent tests where samples were achieved in triplicate. Significance was identified by Tukey’s multiple comparisons test for multiple comparisons. Group differences were considered statistically significant if *** *p* < 0.001 and **** *p* < 0.0001 compared with control, as measured by one-way ANOVA.

**Figure 11 ijms-25-02181-f011:**
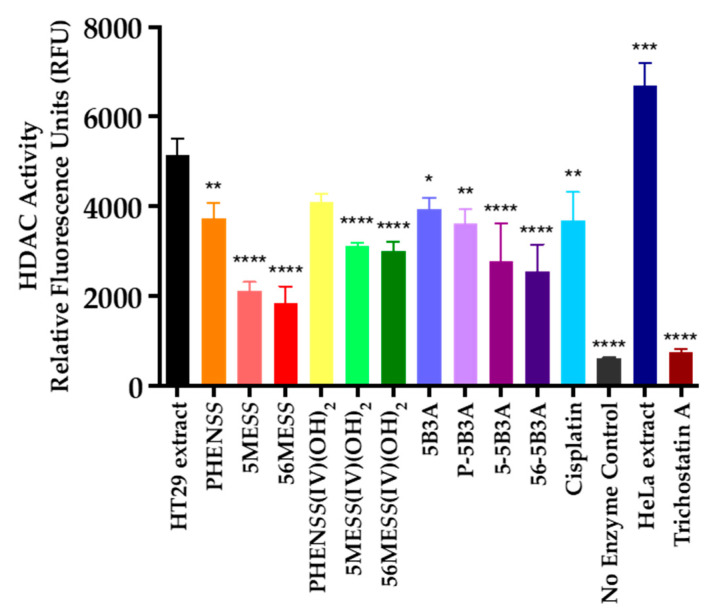
HDAC activity upon treatment with **P-5B3A**, **5-5B3A** and **56-5B3A**, 5B3A, the equivalent platinum(II) precursors (**PHEN*SS***, **5ME*SS*** and **56ME*SS***) and platinum(IV) scaffolds (**PHEN*SS*(IV)(OH)_2_**, **5ME*SS*(IV)(OH)_2_** and **56ME*SS*(IV)(OH)_2_**), and cisplatin at 72 h. Basal fluorescence was measured every min for 60 min. HeLa nuclear extract (ab156064) was used as positive control for activity, Trichostatin A (TSA) as inhibitor control for HDAC activity, and a No enzyme control (kit buffer and developers ab156064). The representative bar graph denotes the RFU for the end point. Data points signify the mean ± SEM. *n* = 3 from three independent experiments where samples were conducted in triplicate. Significance was identified by Tukey’s multiple comparisons test for multiple comparisons. Group differences were considered statistically significant if * *p* < 0.05, ** *p* < 0.01, *** *p* < 0.001 and **** *p* < 0.0001 compared with HT29 extract (control), as measured by one-way ANOVA.

**Table 1 ijms-25-02181-t001:** A summary of the peak maxima in the UV and CD spectra of **P-5B3A**, **5-5B3A** and **56-5B3A**.

Platinum(IV) Complexes	UV λ_max_ nm (ε/M.cm^−1^ ± SD × 10^4^)	CD λ_max_ nm (Δε/M.cm^−1^ × 10^1^)
**P-5B3A**	304 (0.73 ± 3.15), 279 (2.16 ± 1.63), 204 (8.08 ± 1.70)	202 (−463), 203 (−485), 205 (−551), 228 (−58.9), 239 (−99.6), 263 (+31), 269 (+5.15), 274 (+21.3), 285 (−16.5), 313 (−32.2)
**5-5B3A**	310 (1.01 ± 0.73), 286 (3.63 ± 0.33), 204 (13.5 ± 3.61)	202 (−433), 203 (−460), 204 (−490), 230 (−35.1), 240 (−65.1), 257 (+16.9), 263 (+28.6), 270 (+30.7), 286 (−53.5)
**56-5B3A**	315 (0.59 ± 0.94), 291 (2.02 ± 1.78), 250 (1.31 ± 2.53), 206 (8.22 ± 1.75)	202 (−376), 204 (−384), 208 (−425), 230 (−6.55), 235 (−25.4), 239 (+7.34), 248 (+73.4), 257 (+101), 275 (+80.7), 288 (+0.91)

**Table 2 ijms-25-02181-t002:** Calculated log k_w_ values of **P-5B3A**, **5-5B3A** and **56-5B3A**.

Platinum(IV) Complexes	log k_w_ Values
**P-5B3A**	1.82
**5-5B3A**	1.91
**56-5B3A**	2.11

**Table 3 ijms-25-02181-t003:** The approximate time points in min for **P-5B3A**, **5-5B3A** and **56-5B3A** at which 50 and 100% reductions occurs in the presence of AsA.

Platinum(IV) Complexes	Approximate Reduction Time Points (min)	
	T_50%_	T_100%_
**P-5B3A**	10–15	60
**5-5B3A**	5–10	30
**56-5B3**	0–5	10

T_50%_: Time at which 50% of reduction occurs. T_100%_: Time at which 100% of reduction occurs.

**Table 4 ijms-25-02181-t004:** The in vitro cytotoxicity values of 5B3A, **P-5B3A**, **5-5B3A** and **56-5B3A** in multiple human cell lines, expressed as GI_50_ (nM). For comparison, the GI_50_ values of cisplatin, oxaliplatin and carboplatin were also included [59,61,65]. Significant increase in potency was identified by a paired T-test with a two-tailed distribution (*n* = 3). Group differences were considered statistically significant if * *p* < 0.05, ** *p* < 0.01, *** *p* < 0.001 and **** *p* < 0.0001 compared with cisplatin.

GI_50_ (nM)							
Cell Lines	5B3A	P-5B3A	5-5B3A	56-5B3A	Cisplatin	Oxaliplatin	Carboplatin
**HT29**	>50,000 ***	200 ± 27 **	18 ± 0.67 **	4.4 ± 1.6 **	11,300 ± 1900	900 ± 200	>50,000
**U87**	>50,000 ***	1900 ± 150 *	240 ± 23	88 ± 7 *	3800 ± 1100	1800 ± 200	>50,000
**H460**	>50,000 ****	430 ± 50 *	30 ± 3	7.8 ± 2.2 *	900 ± 200	1600 ± 100	14,000 ± 1000
**A431**	>50,000 ****	1200 ± 100 **	93 ± 8 **	33 ± 8.4 **	2400 ± 300	4100 ± 500	24,000 ± 2200
**Du145**	>50,000 ****	190 ± 23 **	10 ± 2 **	1.2 ± 0.6 **	1200 ± 100	2900 ± 400	15,000 ± 1200
**BE2-C**	>50,000 ****	13,000 ± 5000 **	340 ± 40	150 ± 24 **	1900 ± 200	900 ± 200	19,000 ± 1200
**SJ-G2**	>50,000 ****	1300 ± 200	200 ± 40 *	110 ± 20	400 ± 100	3000 ± 1200	5700 ± 200
**MIA**	>50,000 ****	310 ± 26 **	31 ± 3 **	12 ± 3 **	7500 ± 1300	900 ± 200	>50,000
**A2780**	>50,000 **	470 ± 18 **	42 ± 4 **	18 ± 6.1 **	1000 ± 100	160 ± 100	9200 ± 2900
**ADDP**	>50,000 ****	330 ± 8.8 ****	23 ± 0.88 ****	6 ± 3 ****	28,000 ± 1600	800 ± 100	>50,000
**MCF-7**	>50,000 **	5400 ± 2800 **	200 ± 21	68 ± 10 **	6500 ± 800	500 ± 100	>50,000
**MCF10A**	44,000 ± 3000 **	280 ± 39 **	19 ± 1.5 **	3.3 ± 0.9 **	5200 ± 520	*Not determined*	>50,000
**Mean GI_50_**	**49,500 ± 3000**	**2084 ± 703**	**104 ± 12**	**42 ± 7**	**5842 ± 610**	**1463 ± 320**	**32,242 ± 1450**

**Table 5 ijms-25-02181-t005:** The in vitro cytotoxicity values of platinum(II) complexes (**PHEN*SS***, **5ME*SS*** and **56ME*SS***) and their corresponding platinum(IV) scaffolds (**PHEN*SS*(IV)(OH)_2_**, **5ME*SS*(IV)(OH)_2_** and **56ME*SS*(IV)(OH)_2_**), expressed as GI_50_ (nM) [59,61,65]. Significant increase in potency was identified by a paired T-test with a two-tailed distribution (*n* = 3). Group differences were considered statistically significant if * *p* < 0.05, ** *p* < 0.01, and **** *p* < 0.0001 compared with cisplatin (see Table 6 for the GI_50_ values of cisplatin).

GI_50_ (nM)						
**Cell Lines**	**PHENSS**	**5MESS**	56MESS	PHEN*SS*(IV)(OH)_2_	5ME*SS*(IV)(OH)_2_	56ME*SS*(IV)(OH)_2_
**HT29**	160 ± 45 **	33 ± 4 **	10 ± 1.6 **	710 ± 300 **	60 ± 6 **	36 ± 7 **
**U87**	980 ± 270 *	320 ± 26 *	35 ± 6.4 *	4900 ± 610	900 ± 58	190 ± 23 *
**H460**	360 ± 35 *	41 ± 5 *	21 ± 2 *	1700 ± 200	60 ± 5 *	190 ± 150 *
**A431**	480 ± 170 **	120 ± 25 **	29 ± 1 **	4300 ± 530 *	360 ± 58 **	120 ± 22 **
**Du145**	100 ± 38 **	22 ± 3 **	4.6 ± 0.4 **	310 ± 92 **	41 ± 5 **	15 ± 2.6 **
**BE2-C**	380 ± 46 **	270 ± 38 **	59 ± 4 **	3000 ± 530	1400 ± 300	240 ± 22 **
**SJ-G2**	330 ± 66	220 ± 10	66 ± 22 **	1700 ± 350 **	640 ± 70	210 ± 45 *
**MIA**	200 ± 57 **	48 ± 2 **	13 ± 2 **	3400 ± 2200	160 ± 29 **	43 ± 2.5 **
**A2780**	230 ± 30 **	61 ± 10 **	76 ± 57 **	800 ± 84	240 ± 9 **	59 ± 7 **
**ADDP**	190 ± 47 ****	34 ± 2 ****	13 ± 2 ****	1300 ± 350 ****	130 ± 22 ****	170 ± 120 ****
**MCF-7**	1500 ± 500 **	200 ± 12 **	93 ± 44 **	16,000 ± 4500	1200 ± 390 **	480 ± 140 **
**MCF10A**	300 ± 58 **	30 ± 2 **	16 ± 1 **	1700 ± 200	130 ± 19 **	61 ± 7 **
**Mean GI_50_**	**434 ± 110**	**117 ± 12**	**36 ± 10**	**3318 ± 880**	**443 ± 81**	**151 ± 50**

**Table 6 ijms-25-02181-t006:** The SCI of **P-5B3A**, **5-5B3A**, **56-5B3A** and cisplatin in all cancerous cell lines.

Cell Lines	Platinum Complexes			
	P-5B3A	5-5B3A	56-5B3A	Cisplatin
**HT29**	1.40	1.06	0.75	0.46
**U87**	0.15	0.08	0.04	1.37
**MCF-7**	0.05	0.10	0.05	0.80
**H460**	0.65	0.63	0.42	5.78
**A431**	0.23	0.20	0.10	2.17
**Du145**	1.47	1.90	2.75	4.33
**BE2-C**	0.02	0.06	0.02	2.74
**SJ-G2**	0.22	0.10	0.03	13.00
**MIA**	0.90	0.61	0.28	0.69
**A2780**	0.60	0.45	0.18	5.20
**ADDP**	0.85	0.83	0.55	0.19

SCI = GI_50_ (normal cancer cell line, MCF10A)/GI_50_ (cancer cell lines).

**Table 7 ijms-25-02181-t007:** ROS production upon treatment with **P-5B3A**, **5-5B3A** and **56-5B3A**, 5B3A, and cisplatin in HT29 colon cells at 24, 48 and 72 h. The RFU values reported for platinum(II) precursors (**PHEN*SS***, **5ME*SS*** and **56ME*SS***) and platinum(IV) scaffolds (**PHEN*SS*(IV)(OH)_2_**, **5ME*SS*(IV)(OH)_2_** and **56ME*SS*(IV)(OH)_2_**) were acquired from the literature for comparison [59,61,65].

Compounds and Platinum Complexes	ROS Production * at Different Time Intervals (RFU)		
	24 h	48 h	72 h
Control	60 ± 3	63 ± 4	85 ± 6
Cisplatin	265 ± 19	299 ± 22	364 ± 18
(Tert-butyl hydroperoxide) TBHP	614 ± 21	477 ± 23	311 ± 18
**PHEN*SS***	174 ± 2	172 ± 9	176 ± 7
**5ME*SS***	204 ± 4	205 ± 3	188 ± 3
**56ME*SS***	240 ± 5	218 ± 3	255 ± 4
**PHEN*SS*(IV)(OH)_2_**	144 ± 5	273 ± 4	303 ± 1
**5ME*SS*(IV)(OH)_2_**	234 ± 1	323 ± 9	335 ± 2
**56ME*SS*(IV)(OH)_2_**	259 ± 3	356 ± 11	438 ± 7
5B3A	138 ± 8	153 ± 6	164 ± 4
**P-5B3A**	156 ± 16	214 ± 12	246 ± 11
**5-5B3A**	160 ± 11	230 ± 9	315 ± 14
**56-5B3A**	274 ± 9	418 ± 15	474 ± 10

* Data are the mean ± SEM of three independent experiments, each performed in triplicate.

**Table 8 ijms-25-02181-t008:** The detected MtMP changes upon treatment with **P-5B3A**, **5-5B3A** and **56-5B3A**, 5B3A, the platinum(II) precursors (**PHEN*SS***, **5ME*SS*** and **56ME*SS***) and the platinum(IV) scaffolds (**PHEN*SS*(IV)(OH)_2_**, **5ME*SS*(IV)(OH)_2_** and **56ME*SS*(IV)(OH)_2_**), and cisplatin in HT29 colon cells at 24, 48 and 72 h.

Compounds and Platinum Complexes	MtMP Changes * at Different Time Intervals (RFU)		
	24 h	48 h	72 h
Control	542 ± 4	507 ± 11	449 ± 10
Cisplatin	232 ± 4	230 ± 4	192 ± 7
Carbonyl cyanide-p-trifluoromethoxyphenylhydrazone (FCCP)	234 ± 3	151 ± 7	84 ± 5
**PHEN*SS***	321 ± 7	305 ± 5	257 ± 6
**5ME*SS***	283 ± 4	261 ± 9	189 ± 11
**56ME*SS***	277 ± 5	211 ± 7	151 ± 7
**PHEN*SS*(IV)(OH)_2_**	328 ± 7	314 ± 6	234 ± 11
**5ME*SS*(IV)(OH)_2_**	311 ± 3	291 ± 9	228 ± 7
**56ME*SS*(IV)(OH)_2_**	291 ± 9	221 ± 5	188 ± 5
5B3A	489 ± 6	415 ± 5	327 ± 9
**P-5B3A**	332 ± 7	317 ± 7	263 ± 5
**5-5B3A**	324 ± 9	280 ± 12	184 ± 4
**56-5B3A**	321 ± 11	267 ± 8	148 ± 7

* Data are the mean ± SEM of three independent experiments, each performed in triplicate.

**Table 9 ijms-25-02181-t009:** HDAC activity inhibition upon treatment with **P-5B3A**, **5-5B3A** and **56-5B3A**, 5B3A, the platinum(II) precursors (**PHEN*SS***, **5ME*SS*** and **56ME*SS***) and platinum(IV) scaffolds (**PHEN*SS*(IV)(OH)_2_**, **5ME*SS*(IV)(OH)_2_** and **56ME*SS*(IV)(OH)_2_**), and cisplatin in HT29 colon cells at 72 h.

Compounds and Complexes	HDAC Inhibition * at 72 h
HT29 extract	5154 ± 40
**PHEN*SS***	3733 ± 37
**5ME*SS***	2124 ± 21
**56ME*SS***	1847 ± 42
**PHEN*SS*(IV)(OH)_2_**	4107 ± 19
**5ME*SS*(IV)(OH)_2_**	3119 ± 9
**56ME*SS*(IV)(OH)_2_**	3011 ± 23
5B3A	3942 ± 28
**P-5B3A**	3620 ± 35
**5-5B3A**	2791 ± 72
**56-5B3A**	2550 ± 57
Cisplatin	3698 ± 59
No Enzyme Control	617 ± 3
HeLa Nuclear Extract	6697 ± 55
Trichostatin A (TSA)	749 ± 8

* Data are the mean ± SEM of three independent experiments, each performed in triplicate.

## Data Availability

All data relevant to the publication are included.

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
