# Peer review of "Platinum(IV) Prodrugs Incorporating an Indole-Based Derivative, 5-Benzyloxyindole-3-Acetic Acid in the Axial Position Exhibit Prominent Anticancer Activity"

_ijms, 2024, doi:10.3390/ijms25042181_

Round 1

Reviewer 1 Report

Comments and Suggestions for Authors The article is, in my opinion, suitable for publication after minor revision because it presents the synthesis of new platinum complexes, although using a well established approach, and their full characterization, especially spectroscopic characterization, which is described in detail in the discussion. The stability of the compounds in pseudo-physiological media and the evaluation of LogP, two significant parameters for judging the pre-requisites of a potential drug, have been determined. The biological experiments have been conducted by assessing the cytotoxicity on a panel of cancer cell lines, and also targeted mechanistic experiments have been carried out.

The authors report a comprehensive work (synthesis, characterization, biological investigation) on platinum(IV) complexes functionalized with an indole skeleton designed for anticancer activity. The paper is well written, and the results are discussed in an organized manner, therefore I recommend its publication after revising some minor points.

1)      I would report, for comparison, the LogKw values of chemotherapic platinum complexes, at least cisplatin

Additional information may be added to the Introduction regarding literature metal complexes containing the indole motif investigated as potential anticancer drugs, and some comparison may be traced with the results presented herein (see for instance - DOI codes - 10.1016/j.ejmech.2020.112506, 10.1016/j.cbi.2023.110742, 10.3390/ijms24010854)

Comments on the Quality of English Language

----

Reviewer 2 Report

Comments and Suggestions for Authors

The authors describe the synthesis and characterisation of three Pt(iv) complexes based on oxaliplatin derivatives that substituted the oxalate leaving group with 1,10-phenantroline and derivatives thereof with one or two methyl groups in position 5 and 6, respectively. The axial positions were unsymmetrically derived with 5-benzyloxyindole-3-acetic acid (5B3A) and a free hydroxy group. The synthesized compounds are well characterized by NMR, UV and CD spectroscopy, mass spectrometry and purity was confirmed by HPLC-UV/Vis and elemental analysis. The activity of the compounds were investigated and compared to cis-, carbo- and oxaliplatin via MTT assays. The mode of action of the drugs was investigated by ROS formation, mytochrondial dysfunction and HDAC inhibition studies.

The authors did a lot of work to investigate the stability, the activity and the to shine light on the mode of action of the complexes. However, the manuscript is overloaded with unneccessary data and should definitely be immensively shortened, starting with the introduction section (e.g. p. 2, line 57-66 can be deleted as not of any use for the reader).

The results and discussion part should just contain valuable information that is presented in a precise manner, which is not the case in this manuscipt:

The synthesis part mainly refers to literature and just summarizes were to find spectra and data (can be summarized in 3 sentences in the material and methods part). Please just explain the novel parts of the synthesis. The counterions are not shown in Figures 4+5 and it is just explained on page 8 (elemental analysis) that the counter ions are nitrates. Please add this information at the necessary positions. Most of the data of table 1 is described in the supporting information, not of further use for the reader and can be deleted. In case of the m/z values: The authors are using a qTOF setup, why are two decimal places presented instead of four? In order to prove the identity every formerly undescribed compound should be investigated via high-resolution mass spectrometry (data should be readily available). Table 2 is also described in the experimental part, therefore not of further use for the reader and should be deleted. Figure 6 is valuable and well explained but the explaining part should definitely be shortened to one significant (set of) signal(s), not all.

Chapter 2.2.1 explains how the data is evaluated which belongs to the materials and methods section, please delete and (briefly!) discuss the results.

Chapter 2.2.2. shows preliminary reduction measurements. The introduction to this section (p11 l314-322) must be deleted as it is also mostly explained in the introduction section.

Chapter 2.2.3: The interested reader from the field of medicinal chemistry might know that hydrolysis is a reaction of a molecule with water. Please shorten. At which wavelength are graphs in S40+S41 recorded? The experimental section says 214 nm or 254 nm. Why was no T0 included and which concentrations were used? Please add information to the experimental part.

Chapter 2.3.1: What were the positive controls in the experiments (not the values from literature)? The authors should at least use oxaliplatin as positive control as this drug is the chemically most similar drug compared to the synthesized compounds. Just citing the values of old experiments is not acceptable as it does not indicate any experimental problems during the assay. Therefore the experimental results are not reliable. It is sufficient to include oxaliplatin and compare the Pt(iv) complexes to its parent compounds (or at least just to discuss the results against oxaliplatin). Shorten this part drastically and repeat the experiments with oxaliplatin (or its Pt(iv) complex with 5B3A to rule out the role of 5B3A if feasible) as positive control.

In table 8 the SCIs are calculated showing that the complexes are in general less selective than cisplatin to cancer cells. Why did the authors include cisplatin and not oxaliplatin as your reference? These compounds are chemically more related to oxaliplatin than cisplatin and this comparison would be considerably more interesting. Please explain. In general the authors should give the results in the tables and discuss them in a brief and understable way what that means for the compounds. (B53A instead of B5A3 in lines 479 and 481)

Chapter 2.3.2: p 16, l. 511- 537 the authors should explain why the 5B3A ligand is used in the introduction (briefly!).  Why did the authors compare their drugs against cisplatin and not oxaliplatin as most related compound?

Chapter 2.3.3: Please again Introduction part to Introduction section. The reduced amount of ROS produced might also be dependend on the cytostatic effect of the drugs which is not mentioned in the text (after 72h ~1/2 of the untreated control which can be referred to the GI50 values).

Chapter 2.3.4: again cut l. 652-680 and transfer to the introduction part to explain why the indole-carboxylic acid pharmacophore is used. Figure 11 is overloaded as nobody is able to see any graphs except HeLA extract and HT29 extract. Anyway why is there also a graph for the HeLa nuclear extract? Was the axial ligand introduced as competitive HDAC inhibitor? What do the author mean with "inhibit the cytoskeleton"?

Materials and Methods part:

Please insert all the relevenat informations in the manuscript and not into the supporting information. Include the synthetic routs about the former not described compounds and give all the analytical information. Include all procedures for your physico- and biochemical investigations and just put all the spectra into the supporting information.

Conclusions

The definition of prodrug is that the active drug is masked into a pharmacologically inactive compound and released upon a specific trigger event. Just the conversion of a Pt(iv) species into a Pt(ii) species is not an indication that your compound is a prodrug. The authors could show that the Pt(iv) species is released upon addition of ascorbic acid, but also that the methylated Pt(iv) complexes are more active than its Pt(ii) core structures. Therefore the definition of a prodrug is just appliable to P5B3A->PHENSS (see Table 6 + 7). Furthermore you are again comparing to cisplatin, but you should compare the biological activity against oxaliplatin.

All in all, the amount of work done by the authors for this publication is impressive and should definitely be considered for publication when a positive control in the biological experiments is included to produce reliable results. The presentation of the results must be improved and the whole manuscript must be shortened(!). It has to be focused on important and specific results that shall be presented in a on-the-point manner. The results and discussion part must not contain experimental descriptions and no information that should be in the introduction. The materials and methods part should contain any relevant information, the supporting information should contain the spectra and not the relevant experimental procedures and setups.

Reviewer 3 Report

Comments and Suggestions for Authors

This work describes the results on the development of three platinum(IV) complexes that incorporate 5-benzyloxyindole-3-acetic acid, a bioactive ligand that integrates an indole pharmacophore, the study of their cytotoxicity and the identification of potential pathways of interaction with cells. At present, due to the high relevance of research on the search for effective and safe substances to combat cancer, there is a fairly wide range of works on relevant topics. This work may also be of interest to readers. The methods of testing the cytotoxicity of the obtained complexes and their precursors, as well as methods of evaluating the effect of these complexes on cells described in this work are generally accepted in this kind of research.  The structure of the obtained complexes was characterized using spectral methods. There are a number of suggestions for improvement of the manuscript:

Major

1) Taking into account the high quartile and impact factor of the IJMS journal it is necessary to supplement the characterization of the obtained complexes with the data of X-ray structural analysis.

2) The authors should clarify the effect of decreased cytotoxicity towards some cell types of some Pt(IV) complexes in comparison with their precursors.

3) According to Tables 6 and 7, the cytotoxicity of almost all Pt(IV) target complexes and their precursors was significantly higher than that of cisplatin. Whereas the level of ROS generated under the action of cisplatin is at or higher than that of the complexes or their precursors (exception 56-5B3A). The authors should provide an explanation for the observed effect.

4) According to Table 10, precursors of Pt(IV) complexes caused a more pronounced impairment of mitochondrial function than the target complexes. The authors should explain this effect and make an assumption about really possible mechanisms of action of Pt(IV) target complexes and the role of Pt(IV) in them.

Minor

1) Authors should indicate the solvent for the data in Table 1.

2) Figures 6 and 7. Authors should increase the font in the axis captions.

3) The abbreviations SCI and MtMP are only deciphered in SI. It is required to decipher them the first time they are mentioned in the paper.

4) The authors should describe the parameter MtMP and its inhibition mechanism in more detail.

5) The authors should clarify the term "poor pharmacokinetics" of HDACis

6) The authors should eliminate unnecessary self-citation.

Round 2

Reviewer 2 Report

Comments and Suggestions for Authors

Thank you very much for the revised manuscript. Most of the remarks were addressed.

minor remarks:

Table 2. last row, the A is missing in 56-5B3

l. 784: [M-H]+ is not a cation, should be [M-H]- (if spectrum was measured in negative polarity); calculated monoisotopic mass is 785.2421, not 785.2400.

l. 797f: [M-H]+ same as l.784; m/z=799.2577, not 799.2560; experimental: m/z = 799.4457.

l.812f: see above: m/z = 813.2734, not 813.2700

major remarks:

All mass spectra provided are far off the usual 5 ppm from calculated values to confirm the identity of the synthesized compounds.

P-5B3A (114 ppm), 5-5B3A (235 ppm), 56-5B3A (197 ppm) (regarding the most abundant molecular peak from the isotopic pattern calculation, not the highest peak in the spectra). Additionally the isotopic pattern of the compounds do not fit to the calculated spectra (e.g. see https://mstools.epfl.ch/info/ using C35H36N5O4Pt)). Please remeasure the HRMS (maybe dilute the sample to lower concentrations, sometimes collapse of the isotopic pattern and shift of the monoisotopic masses are related to an overloaded mass detector due to high concentrations/ionzation of the sample).

Reviewer 3 Report

Comments and Suggestions for Authors

The authors have tried to take into account all the reviewer's comments and suggestions. The manuscript can be recommended for publication.

Author Response

We would like to thank the reviewer for taking the necessary time and effort to review the manuscript. We sincerely appreciate your valuable comments and suggestions, which helped us in improving the quality of our manuscript.

Round 3

Reviewer 2 Report

Comments and Suggestions for Authors

Thank you very much for the revised manuscript. First in difference to my former remarks the [M-H] ions will of couse be cationic. Sorry for the mistake.

Unfortunately the authors stated that the hrms data is not usable due to an most probably uncalibrated mass spectrometer. Though the mass spectra are somehow off (the used structures in SI are missing the nitrogens of the phenanthroline rings and are taken for carbons instead), the compounds are well enough characterized by NMR spectroscopy. Please just state the ESI masses in the experimental part (either integer or with 2 digits, not HRMS, e.g. for P-5B3A: ESI-MS: calculated for [M-H]+: m/z = 813.27; experimental: m/z = 813.37) and delete the spectra from the supporting information.
